# DrivAerML: High-Fidelity Computational Fluid Dynamics Dataset for Road-Car External Aerodynamics

## Abstract

Machine Learning (ML) has the potential to revolutionise the field of automotive aerodynamics, enabling split-second flow predictions early in the design process. However, the lack of open-source training data for realistic road cars, using high-fidelity CFD methods, represents a barrier to their development. To address this, a high-fidelity open-source (CC-BY-SA) public dataset for automotive aerodynamics has been generated, based on 500 parametrically morphed variants of the widely-used DrivAer notchback generic vehicle. Mesh generation and scale-resolving CFD was executed using consistent and validated automatic workflows representative of the industrial state-of-the-art. Geometries and rich aerodynamic data are published in open-source formats. To our knowledge, this is the first large, public-domain dataset for complex automotive configurations generated using high-fidelity CFD.

## Introduction

External aerodynamics plays an important role in the design of road vehicles (30). Aerodynamic drag directly influences emissions from internal combustion engines, and the range of electric vehicles. For high-performance cars, the magnitude and distribution of aerodynamic downforce are key factors in cornering and handling (37). Pressure fluctuations from turbulent airflow can give rise to aeroacoustic noise sources, with a negative impact on passenger comfort (53). Especially in the consumer vehicle segment, these factors are decisive differentiators influencing product competitiveness. Alongside aesthetic design, aerodynamics is therefore central in defining the external shape of the vehicle.

Engineering approaches to evaluate the external aerodynamics and aeroacoustics of road vehicles began in the 1960s with wind tunnels, which remain an important tool to this day (64). Although an approximation of true open-road conditions, wind tunnels have the advantage of measuring real airflow over real vehicles (often at full scale) (31). Furthermore, variations of vehicle configurations (e.g. wheel and tyre options, ride height) and operating conditions (e.g. vehicle speed, yaw angle) can be assessed very efficiently. Alongside forces, modern techniques also allow measurement of surface pressure and off-body quantities at selected locations (61).

Computational Fluid Dynamics (CFD) is a more recent innovation, routinely used since the 1990s (30). CFD can simulate real open-road conditions and provide insight into the entire flow field. However, each configuration change requires a separate simulation, so CFD is typically used only for a subset of scenarios measured in the wind tunnel. Furthermore, computational cost necessitates modelling approximations, most notably regarding the onset and effects of turbulence.

Strategies for turbulence modelling in CFD (59) occupy different positions in the trade-off between computational cost and accuracy. Reynolds-averaged Navier-Stokes (RANS) approaches predict the time-averaged flow field using statistical turbulence models. They are inexpensive and run within a few hours, however the correlation to measurements is unreliable for complex flows (e.g. large-scale flow separation) (9; 4; 5; 8).

Greater accuracy is achieved by resolving some of the turbulent motion using "scale-resolving simulation" (SRS) methods. Depending on the method, these time-accurate simulations can easily consume an order of magnitude greater computational resources than RANS, due to the wide range of length scales inherent to turbulence (3). A road vehicle is metres long and multiple-second

time samples are required for robust statistics. Resolving all turbulent eddies via Direct Numerical Simulation (DNS), or only the largest local scales via wall-resolved Large-Eddy Simulation (WRLES), will remain unaffordable for decades to come, since the eddies are extremely small close to the vehicle surface at full-scale Reynolds number.

Bridging the inner boundary layer region using wall-modelled LES (WMLES) (39) delivers a significant cost reduction. However the eddies in the outer boundary layer still need to be resolved, and towards the front of the vehicle the boundary layers are thin (in the order of mm). Pioneering WMLES have been demonstrated e.g. by participants of the Automotive CFD Prediction Workshop series (33), but the approach is not yet routinely applied in industry.

Hybrid RANS-LES (HRLES) methods (such as Delayed Detached-Eddy Simulation (DDES) (58)) are the most mature and widely used category of SRS. These generally model attached boundary layers entirely with RANS, deploying the scale-resolving capability of LES only in separated flow where the eddies are larger. Typical grids for full scale vehicles have minimum cell spacings in mm, solved with time steps in tenths of milliseconds (33). HRLES is currently applied routinely in the automotive industry as a higher-fidelity option alongside RANS (4; 33). Simulation turnaround times between several hours (overnight) and a few days are typical.

## MACHINE LEARNING FOR AUTOMOTIVE AERODYNAMICS

Machine Learning (ML) has the potential to revolutionise automotive aerodynamics by offering faster and cheaper ways to predict fluid flow. For example, surrogate ML models, once trained, can be used to predict the aerodynamic performance of a given geometry in seconds, rather than hours or days required by CFD. This can be used to give vehicle designers near-instant aerodynamic feedback, as well as to accelerate design optimisation studies (2). The development of ML for fluid dynamics has seen significant progress in recent years, and a review is beyond the scope of this paper (see e.g. the review of Lino et al. (45)). Generally, progress has been achieved in two key directions, namely physics-driven and data-driven approaches. Physics-driven approaches often include the partial differential equation (PDE) to the loss function in methods such as Physics-Informed Neural Networks (PINNs) (57) and Physics-Informed Neural Operators (PINOs) (43).

In contrast, data-driven approaches do not explicitly learn a PDE but rather use simulation data to learn the solutions or solution operator (16), e.g. supervised learning to minimise the difference between the predicted and "true" solutions. Examples include message-passing graph neural networks (GNNs) (10; 25), e.g. MeshGraphNets (56; 17; 38) and pure data-driven Neural Operators (40; 41; 27; 42)

The size and quality of training datasets are of paramount importance to the development of any ML method (especially data-driven approaches). Canonical flows (e.g. boundary layers, vortices, wakes) are already challenging to predict in isolation. In industrial aerodynamics, multiple such features interact in highly non-linear ways, raising the challenge to a higher level. It is therefore doubtful that an ML approach trained only using canonical flows can succeed in complex scenarios. For the same reason, it is important to include complex cases when *testing* ML approaches intended for such applications. For traditional CFD, the ability of a single model to predict a wide range of different flows (without the need for ad-hoc tuning) is a hallmark of a general-purpose approach.

## RELATED WORK

The presented dataset, named "DrivAerML", is not the first aimed at ML in automotive aerodynamics. Jacobs et al. (34) demonstrated the potential for surrogate ML models to speed up automotive design using a dataset of 1000 DrivAer variants but the data has never been made publicly available. The recent "DrivAerNet" dataset of Elrefaie et al. (15), which was generated concurrently to DrivAerML and is also open-source (CC-BY-NC), features an impressive 4000 DrivAer variants. Since the CFD data was generated with RANS on comparatively coarse meshes (8M to 16M cells), it is inherently in the low-fidelity category.

The DrivAerML dataset has been prepared in conjunction with two further high-fidelity CFD datasets (i.e. deploying SRS methods), for the simpler Ahmed ("AhmedML") (7) and Windsor ("WindsorML") (6) car bodies. The consistency of file structure and naming conventions among these datasets is

intended to facilitate the development and testing of ML approaches across multiple datasets, ensuring better robustness and generalisability.

### OBJECTIVES AND MAIN CONTRIBUTIONS

Progress in ML for aerodynamics is constrained by the scarcity and expense of high-quality data, and it is the primary motivation of this work to address this. The paper describes the creation and publication of the DrivAerML dataset, as well as the validation of the deployed CFD methods. The DrivAerML dataset has the following specifications:

- A complex geometry relevant to the field, for which the "OCDA" variant (32) of the established DrivAer (29; 28) research model has been selected as baseline[1].
- A large dataset consisting of 500 variants of the baseline geometry, covering the main features seen on this category of road vehicle.
- The use of hybrid RANS-LES, the highest-fidelity scale-resolving CFD approach routinely deployed by the automotive industry, ensuring best possible correlation to experimental data.
- Consistency of the dataset, ensured by automated meshing and simulation workflows with statistical quality control.
- A rich dataset, comprising full flow-field data, surface data and application-relevant quantities (e.g. force and moment coefficients).
- An open-source dataset, freely available, free of usage constraints and using open-source file formats.

Based on input from automotive companies, we decided to focus only on geometry variants of the DrivAer vehicle rather than also changing the boundary conditions (e.g. Reynolds number or incoming flow angle).

To the best of our knowledge, the DrivAerML dataset represents the first large, open-source ML training dataset comprising high-fidelity CFD data for complex automotive aerodynamics geometries. Although it primarily targets data-driven surrogate ML approaches, the dataset may also prove useful for other ML approaches, or even for purposes beyond the field of ML entirely.

The paper is organised as follows: The baseline case and experimental data is described in Sect. , and the generation of the high-fidelity CFD dataset is covered in Sect. , including descriptions of the methods and their validation. The contents and structure of the dataset are then described in Sect. before discussing conclusions and limitations of the current work in Sect. . Extensive additional material is provided in the Supplementary Information; SI.

### BASELINE GEOMETRY AND FLOW CONDITIONS

The DrivAer model was introduced (29; 28) as an open-source road vehicle for research into automotive aerodynamics, giving a more realistic stepping stone from simpler geometries such as the Ahmed (1) and Windsor (62; 54) bodies. A variant including cooling flow was defined by Ford (32). With three defined configurations—estate, fastback and notchback—it has been subject to numerous experimental studies (29; 65; 12; 62; 31) as well as a wide range of CFD investigations (28; 26; 65; 55; 5; 12; 35; 36; 44; 11; 14; 34; 33; 9).

For the baseline geometry of the DrivAerML dataset, we adopt the specific test case from the 2nd to 4th Automotive CFD Prediction ("AutoCFD") workshops[2], to benefit from and build upon this extensive body of work (33). The geometry is the notchback variant of the Ford OCDA DrivAer (32) with static wheels, sealed cooling inlets and a detailed underbody, Fig. 1a. Although wheel rotation effects are important, static wheels were chosen in the AutoCFD workshops to reduce complexity, since the focus was on comparing CFD approaches for the overall flow field.

---

[1]This Ford-designed variant was introduced with an optional engine bay cooling path, hence the nomenclature "Open-Cooling DrivAer" (OCDA). The cooling path is closed in the considered geometry, hence the differences to the original DrivAer model are minor, see Hupertz et al. (32).

[2]https://autocfd.org

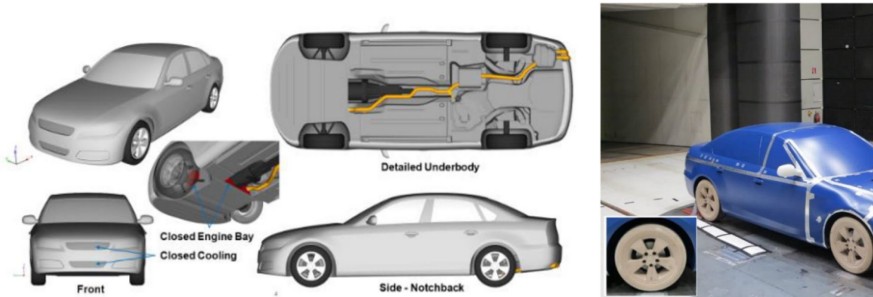

(a) Details of geometry                    (b) 1:1 scale model in wind tunnel

Figure 1: Ford OCDA DrivAer model with closed cooling inserts (33).

The CFD setup assumes incompressible flow with a freestream velocity of $U_\infty = 38.889$ m/s, ambient temperature of $T = 293.15$ K and kinematic viscosity of $\nu = 1.507 \times 10^{-5}$ m$^2$/s. The Reynolds number of $Re_L = U_\infty L/\nu = 7.19 \times 10^6$ based on the wheelbase $L = 2.786$ m is large enough to assume turbulent flow over most of the car. The reference frontal area $A = 2.17$ m$^2$ is used for force and moment coefficients.

A large "open road" domain is used in the CFD, whereas the Pininfarina wind tunnel used in the Ford experiments (31) has an open-jet test section with 11 m$^2$ nozzle area, Fig. 1b. To mimic the wind tunnel ground boundary layer, its starting point in the CFD is located at $x_{\mathrm{BL}} = -2.339$ m $\approx -0.84\,L$, 2.346 m upstream of the front axle. Symmetry conditions are set for the lateral and top boundaries. Due to the large domain, the blockage ratio is negligible (approx. 0.25%) in all simulations.

## GENERATION OF HIGH-FIDELITY CFD TRAINING DATASET

### PARAMETRISATION AND GENERATION OF GEOMETRY VARIANTS

A set of morphing boxes was constructed around the baseline geometry using the ANSA software of BETA-CAE Systems[3], see Fig. 25a, allowing geometry variants to be created in a systematic manner. Figs. 25b-25d show the 16 morphing parameters and their range constraints, designed to avoid unrealistic shapes based on engineering judgement. The wide range of geometries is intended to produce different flow topologies and to test the generalisability of ML approaches.

A design of experiments (DoE) tool in ANSA was used to create the parametric values for 500 experiments using a Modified Extensible Lattice Sequence algorithm, which fills the parameter space evenly, also for subsets of and extensions to the dataset. The DoE and morphing process was automated through Python scripts to save each of the 500 variants along with metrics such as wheelbase and frontal area. The choice of 500 geometries was based upon a mixture of computational budget as well as expectations of industrial feasibility. The dataset could be expanded in future work, based on user feedback.

### COMPUTATIONAL MESHES

The models were meshed in ANSA version 24.1.0 using the HeXtreme algorithm, which generates hexa-dominant & polyhedral meshes. Approximately 160 million cells were generated in less than an hour and satisfied OpenFOAM quality criteria. The boundary layer mesh was optimised for wall functions, with 7 layers, a first layer height of 0.75 mm, a total layer height of 12 mm and a variable growth rate between 1.2 and 1.4. The choice of high $y^+$ mesh vs. resolving to the wall was based on findings from the 2nd AutoCFD workshop, where direct comparisons by multiple participants found only a minor difference in results (33), with benefits of significantly faster run times. The mesh in various sensitive areas, such as the wake, the underbody and the mirrors, was refined using size fields that follow the geometry and extend downstream to capture these features, see Fig. 3. The overall mesh resolution and topology was refined based on industrial feedback during the AutoCFD

---

[3]https://www.beta-cae.com/ansa.htm

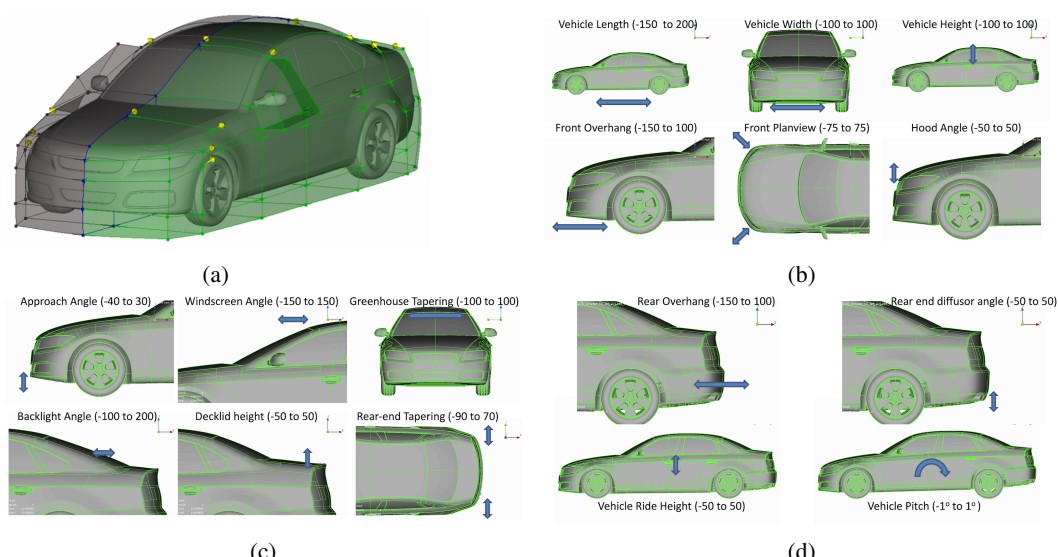

Figure 2: Generation of geometry variants based on the baseline DrivAer model. Visualisation of ANSA morphing boxes (a), visualisation of the 16 design parameters (b-d).

workshop series. The aim is to represent current industrial SRS meshing practice, hence a grid refinement study is not undertaken. The whole meshing process was automated using Python scripts.

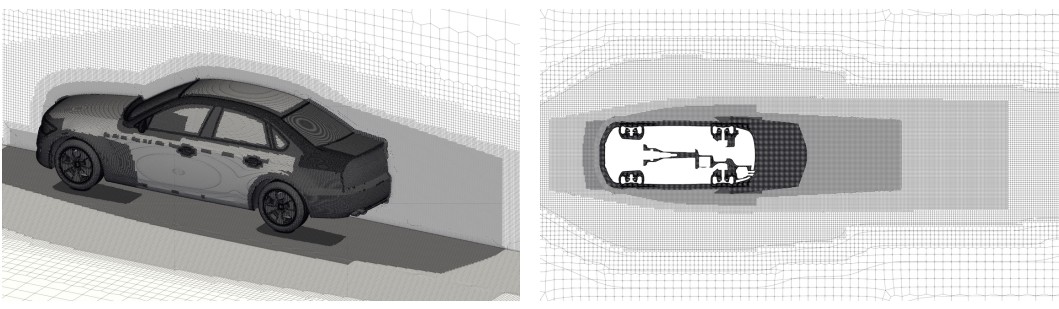

(a) Car surface, ground and $y = 0$ plane        (b) $z$-normal slice through wheel axles

Figure 3: Mesh visualisations for the DrivAer baseline geometry.

## CFD METHODS, WORKFLOW AND VALIDATION

The open-source software library OpenFOAM (v2212) was used to solve the incompressible Navier–Stokes equations via the finite volume method (FVM) (63). The incompressible (constant density) assumption is valid for such low Mach numbers. The high Reynolds number requires a turbulence model (see Sect. ), for which a scale-resolving HRLES approach is chosen as the highest fidelity level routinely applied in the automotive industry. The model of Fuchs, Mockett et al. (47; 23; 22) is used, which a variant of the Delayed Detached-Eddy Simulation (DDES) approach of Spalart et al. (58). It uses the Spalart-Allmaras RANS model (60) in the near-wall region and an LES formulation equivalent to the $\sigma$ model of Nicoud et al. (52) in regions of separated flow. The latter accelerates the transition from RANS to LES after separation, a problem commonly referred to as the "grey area" (49). The approach has been successfully applied for numerous applications and in different CFD codes (20; 18; 21; 48; 19; 24). To ensure that RANS is active throughout the entire boundary layer, the "enhanced protection" (EP) shielding formulation from Deck & Renard's ZDES approach (13) is additionally applied. This avoids the problem of modelled stress depletion (46) from unwanted LES-mode activity inside attached boundary layers, which can induce premature flow separation where the near-wall grid is fine. See SI for a more detailed description of the simulation approach.

**CFD workflow:** The fully-automated simulation workflow, is illustrated in Fig. 4. Taking the mesh as input, the workflow executes domain decomposition, parallel flow simulation and post-processing. Validated, application-specific best practices are automatically applied alongside computational benchmarking data to optimise high-performance computing (HPC) efficiency. The automated workflow ensures setup consistency between simulations.

Achieving *statistical* consistency is challenging, since the simulation times needed to bridge the initial transient (memory of arbitrary initial conditions) and converge the mean vary widely and unpredictably from case to case (50). Simulation runtime is dynamically controlled by interfacing the solver with the time series analysis tool Meancalc, which detects initial transient and quantifies statistical error (51). The simulation stopping criteria were:

1. Initial transient, detected using the drag, lift and side forces, is bridged;
2. Target statistical accuracy of $\pm 1.5$ drag counts[4] is achieved.

This approach ensures that all data is free of initial transient and meets defined statistical accuracy targets whilst optimising HPC cost. The flow field averaging sample is subsequently trimmed to the optimal transient-free length using a custom OpenFOAM utility, ensuring statistical consistency with the mean integral forces (see SI for more details).

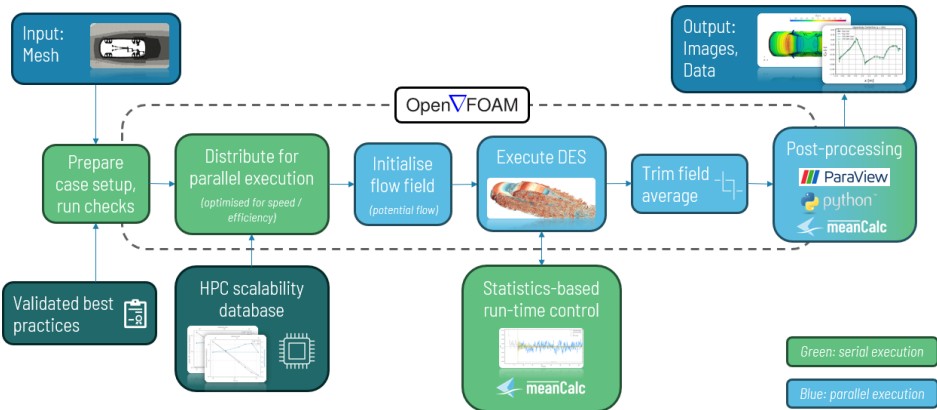

Figure 4: Overview of automated simulation workflow.

All simulations were run on Amazon Web Services (AWS) using a dynamic HPC cluster provisioned by AWS ParallelCluster v3.7.2. A typical simulation took around 40 hours on 1536 cores using Amazon EC2 hpc6a.48xlarge instances (AMD Milan-based processor with 96 cores per node), with variation due to mesh size and statistical convergence. The OpenFOAM code was run in double precision mode using IntelMPI for parallel communication between cores.

**Validation of numerical methodology:** The CFD methodology has been validated against experimental data of Hupertz et al. (31) for the baseline geometry ("case 2a"), and an additional case ("2b") with added front wheel air deflectors (FWAD) (figures in SI).

Instantaneous vortex structures are shown in Fig. 5a. Rich resolved turbulence is seen in areas of separated flow, resolved by the model's LES mode. In contrast, the attached boundary layers (e.g. on the roof and bonnet) are modelled with RANS and generally free of resolved turbulence, except for isolated pockets shed from the windscreen cowl. The fine-grained eddies and absence of spurious oscillations indicate the numerical setup's low-dissipation and robustness, respectively.

Experiment and CFD are generally in close agreement for the time-averaged pressure coefficient over the upperbody centreline (Fig. 5b), with negligible influence of the FWAD seen here. The constant component of the offset between CFD and experiment is due to minor differences in the reference pressure. The rear notch at $x \approx 3.25\,\mathrm{m}$ is a sensitive area, with strong scatter typical in CFD (33). The measured pressure plateau, indicating a small recirculation region, is less pronounced in the CFD.

---

[4]An automotive drag count corresponds to a change in drag coefficient of 0.001

However, visualisation of the CFD surface flow (see SI) reveals a symmetric pair of recirculation regions at the notch, either side of the nominally attached centreline.

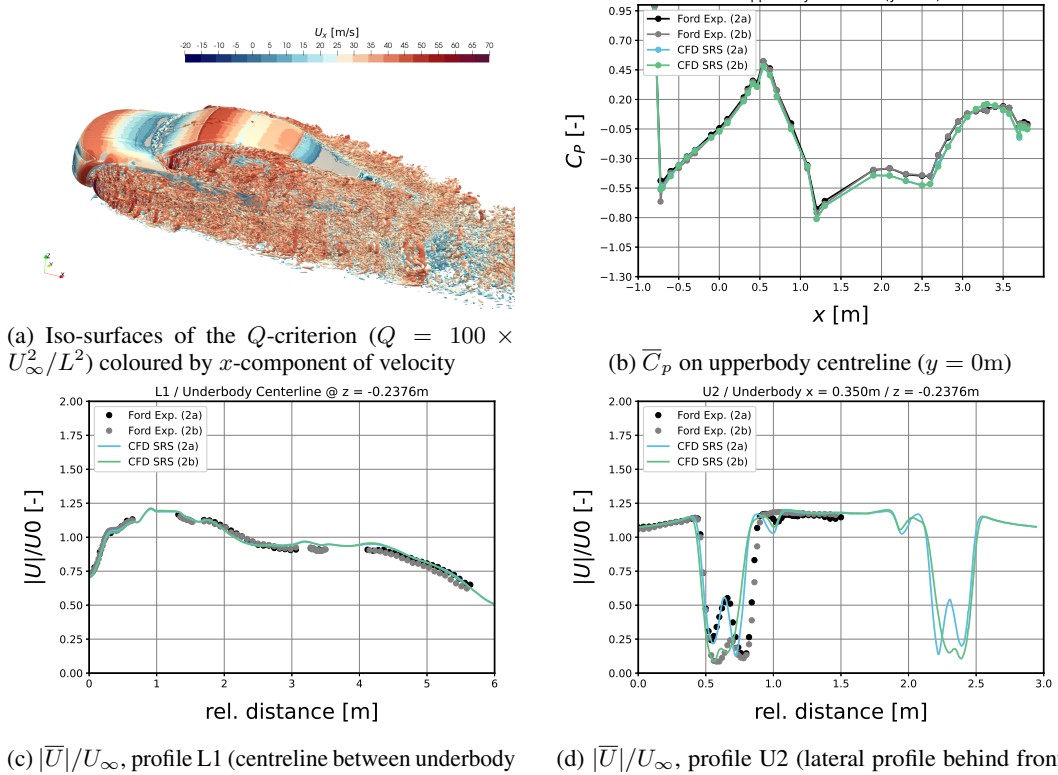

(a) Iso-surfaces of the $Q$-criterion ($Q = 100 \times U_\infty^2/L^2$) coloured by $x$-component of velocity

(b) $\overline{C}_p$ on upperbody centreline ($y = 0$m)

(c) $|\overline{U}|/U_\infty$, profile L1 (centreline between underbody and ground, $z = -0.2376$ m)

(d) $|\overline{U}|/U_\infty$, profile U2 (lateral profile behind front wheels, $x = 0.35$ m, $z = -0.2376$ m)

Figure 5: Comparison of local flow quantities for cases 2a and 2b with Ford experimental data from Hupertz et al. (31).

The airspeed under the vehicle is important for the magnitude and distribution of aerodynamic downforce. Fig. 5c shows encouraging agreement between CFD and experiment along the underbody centreline, another area of strong scatter in CFD (33). The influence of the FWAD is again weak here. In contrast, the front wheel wakes (Fig. 5d) show a clear effect of the FWAD, which is well captured by the CFD. The inboard wheel wake in the CFD appears narrower than in the experiment at this profile location.

The overall conclusion is positive, with encouraging levels of agreement generally seen between experiment and CFD. Where deviations do occur, e.g. in the vertical force and wake velocity profiles, our results agree with the "CFD consensus" from the AutoCFD workshops. It can therefore be concluded with confidence that deviations are due to setup differences (e.g. open-road CFD vs. wind tunnel domains).

## DATASET CONTENTS AND AVAILABILITY

The time-averaged outputs from the simulation of each geometry variant (e.g pressure, velocity & turbulence quantities) were integrated into a dataset structure that maintains consistency with the associated AhmedML (7) and WindsorML datasets (6). A complete list of the dataset contents is given in the SI. For each geometry variant, flow field data for the entire volume (.vtu format), the car surface (.vtp format), 2D slices of the volume in the $x$, $y$ & $z$ directions, geometry data (.stl format), time-averaged force coefficients, and flow visualisation images are provided.

Figures 6c & 6d shows the variation of the lift and drag coefficients for all 500 design variants, showing a variability representative of notchback-type car geometries. There is a significant spread

in drag and lift, due to large changes in flow-field separation, which is illustrated in Figures 30a and 30b for the total pressure coefficient for a high and low drag variant respectively.

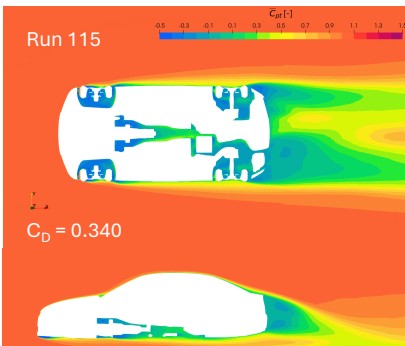

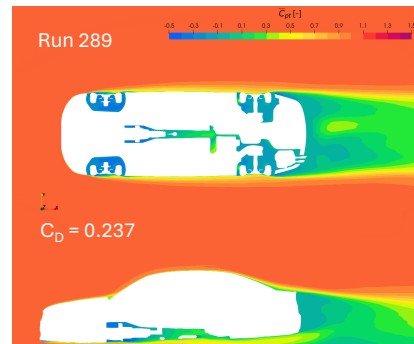

(a) Total pressure coefficient for high drag geometry variant example

(b) Total pressure coefficient for low drag geometry variant example

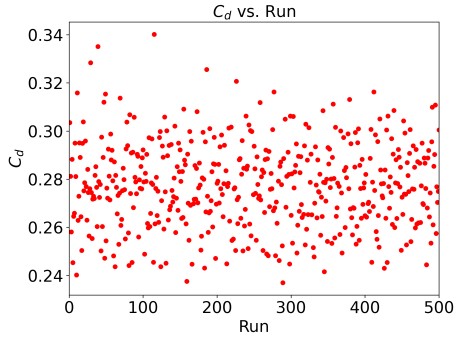

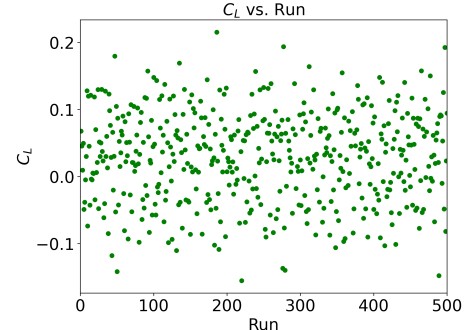

(c) Variation of drag coefficient against run number

(d) Variation of lift coefficient against run number

Figure 6: Variation of total pressure coefficient and force coefficients across the dataset.

The dataset is provided with the CC-BY-SA license[5] (which permits commercial use) and is available to download at no cost through Amazon S3 without the need for an AWS account. Full details are provided in the SI and the dataset README[6].

## ML EVALUATION

We conducted an example ML evaluation using a Graph Neural Network (GNN) approach (more details in the SI) to demonstrate how this dataset could be used to train a ML model to predict unseen cases. We find that using a 60/20/20 split of train, validation and test, it is possible to obtain a MAE of less than 0.035 for the drag coefficient and MAE less than 0.0164 for the lift coefficient via the integration of predicted wall-shear stress and pressure quanities on the 8M node surface mesh of the vehicles (shown in Figure 7). We hope many more groups will use this data to improve upon this and go further to also predict the volume quanities using a range of different ML approaches.

## CONCLUSIONS

A new large scale, public dataset has been established to advance the state of the art for the development of ML methods for the automotive external aerodynamics community. A specific emphasis has been placed on the high fidelity level of the CFD data, using scale-resolving methods, consistent meshing and simulation workflows, and computational grids closely aligned to current industrial best practices. ML models can therefore be trained and tested with greater confidence regarding their accuracy and applicability to complex industrial challenges.

---

[5]https://creativecommons.org/licenses/by-sa/4.0/deed.en
[6]https://xxxxx.s3.us-east-1.amazonaws.com/drivaer/dataset/README.txt

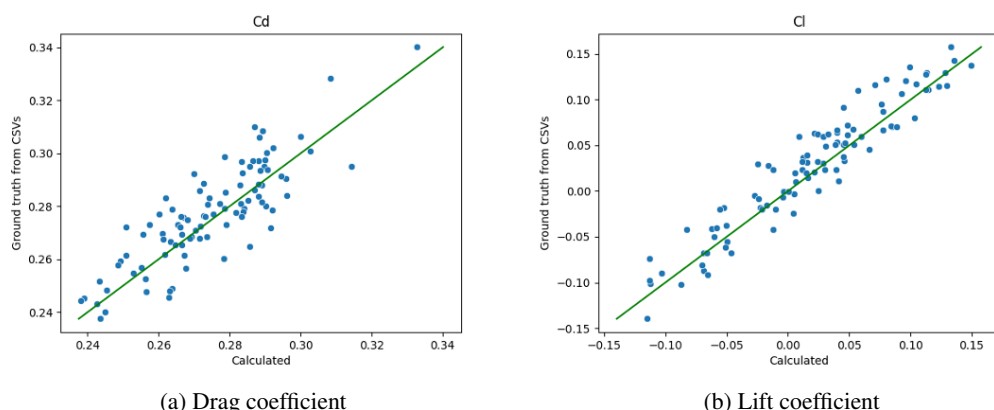

(a) Drag coefficient                    (b) Lift coefficient

Figure 7: Actual vs predicted for the force coefficients obtained through integration of the wall-shear stress and pressure

## LIMITATIONS

Whilst the DrivAerML dataset goes beyond current public-domain datasets in several respects, a number of remaining limitations could be addressed in future work. Parameter variations could be expanded to help build more complete ML models, e.g. with variations of inflow conditions (speed and yaw), the effect of moving ground and rotating wheels, or the inclusion of under-hood cooling flow. Different baseline vehicle geometries could expand the dataset to include e.g. fastback, estate, SUV categories. The inclusion of coarser (and potentially finer) mesh resolutions could facilitate the development of super-resolution and downsampling techniques, as well as addressing the challenges of training ML models on such large mesh counts. The dataset could be extended to include RANS results to investigate transfer learning between fidelity levels, which may lead to more computationally efficient ML approaches.

Whilst there are limitations to this work we believe this is one of the first examples of a large-scale dataset, developed specifically for the automotive community using state-of-the-art CFD approaches. We look forward to learning about other potential uses of the dataset and welcome feedback.

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

CONTENTS

# A   NUMERICAL METHODOLOGY

The open-source software library OpenFOAM (v2212) has been employed in this work for generating the dataset. In this case, the incompressible Navier-Stokes equations for a Newtonian fluid are solved to compute the unsteady fluid motion around the car, where the partial differential equations are discretised in the software package via the finite volume method (FVM) (24). In addition, Reynolds-averaging is applied to the equations to reduce the computational costs of solving. However, this introduces additional modelling empiricism in the form of a turbulence model that is subsequently required (see Sect. A.1). The equations can then be written in Einstein notation and Cartesian coordinates as:

$$\frac{\partial U_i}{\partial x_i} = 0 \,, \tag{1}$$

$$\frac{\partial U_i}{\partial t} + \frac{\partial (U_j U_i)}{\partial x_j} = -\frac{\partial p^*}{\partial x_i} + \frac{\partial}{\partial x_j} \left[ (\nu + \nu_t) \left( \frac{\partial U_i}{\partial x_j} + \frac{\partial U_j}{\partial x_i} \right) \right] \,, \tag{2}$$

where Eq. 1 describes the conservation of mass and Eq. 2 the conservation of momentum. The three-dimensional vector $U_i$ contains the instantaneous velocity components in all three Cartesian directions, $p^* = p/\rho$ is the kinematic pressure with the dimensions $\mathrm{m}^2/\mathrm{s}^2$ and $\nu$ is the molecular kinematic viscosity of the fluid (which is a constant fluid property). The quantity $\nu_t$ is referred to as (turbulent) eddy viscosity and results from the Reynolds-averaging procedure and the Boussinesq linear eddy viscosity assumption that is applied to the Navier-Stokes equations. An additional closure model referred to as a turbulence model is required to compute the $\nu_t$ field, which is described in more detail in Sect. A.1. The assumption of an incompressible fluid (i.e. $\rho = $ const.) is commonly used for automotive applications, since local Mach numbers Ma (i.e. ratio between velocity magnitude $|U_i|$ and speed of sound $c$, $\mathrm{Ma} = |U_i|/c$) are typically low. The Mach number can be interpreted as a measure for compressibility effects. As such a common threshold below which compressibility effects can be neglected is $\mathrm{Ma} = 0.3$. In this study, the highest local velocities are around $70\ \mathrm{m/s}$ so that the maximum local Mach number is approximately $0.2$, which indicates that the compressibility assumption is justified here.

A fundamental scaling parameter in fluid dynamics is the dimensionless Reynolds number Re, which represents the ratio between inertial and viscous forces in a flow. It is defined by a characteristic velocity scale $U_{\mathrm{ref}}$, a characteristic length scale $L_{\mathrm{ref}}$ and the kinematic viscosity $\nu$:

$$\mathrm{Re} = \frac{U_{\mathrm{ref}} L_{\mathrm{ref}}}{\nu} \tag{3}$$

The Reynolds number is often used in fluid dynamics to manage "scale effects", e.g. between a small-scale test model in a wind tunnel and a full scale production vehicle. In case the Reynolds number is equivalent between both applications, the flow is assumed to be statistically equivalent when all flow quantities are normalised accordingly. In addition, the Reynolds number delivers an indication about the flow regime, where a low Reynolds number corresponds to a laminar flow state (where viscous forces are dominant) and a high Reynolds number triggers a turbulent flow state that is dominated by inertial forces. In external automotive applications, the flow is commonly considered to be turbulent in nature since Reynolds numbers are typically very high, so that three-dimensional turbulent eddies of different temporal and spatial sizes are generated in the flow field.

In the following sub-sections, the turbulence modelling approach applied in this work is detailed in Sect. A.1 before the solver and numerical infrastructure is described in Sect. A.2.

## A.1   TURBULENCE MODELLING

The turbulence model employed in this work belongs to the class of hybrid RANS-LES models that is able to switch locally between a lower fidelity RANS model in near-wall regions and a high-fidelity LES model in detached flow regimes. Especially the application of LES to separated flow regions increases the fidelity of the method significantly relative to RANS, since LES allows to resolve most of the turbulent eddies in the flow.

The model used in this work is based on the original delayed detached-eddy simulation (DDES) formulation proposed by Spalart et al. (21), implemented on the basis of the Spalart-Allmaras RANS model (20). A number of key modifications have been applied to this model, which are described in Fuchs et al. (6). The resulting model is referred to as SA-$\sigma$-DDES in this work. It solves for one additional transport equation of the modified eddy viscosity $\tilde{\nu}$:

$$\frac{\partial \tilde{\nu}}{\partial t} + U_j \frac{\partial \tilde{\nu}}{\partial x_j} = P_{\tilde{\nu}} - \epsilon_{\tilde{\nu}} + D_{\tilde{\nu}} \,, \tag{4}$$

where the production ($P_{\tilde{\nu}}$), dissipation ($\epsilon_{\tilde{\nu}}$) and diffusion ($D_{\tilde{\nu}}$) terms of the $\tilde{\nu}$ transport equation are defined via:

$$P_{\tilde{\nu}} = C_{b1} \tilde{S} \tilde{\nu} \,, \qquad \epsilon_{\tilde{\nu}} = C_{w1} f_w \left( \frac{\tilde{\nu}}{L_{\text{DDES}}} \right)^2 \,, \tag{5}$$

$$D_{\tilde{\nu}} = \frac{1}{\sigma_{\tilde{\nu}}} \left[ \frac{\partial}{\partial x_j} \left( (\nu + \tilde{\nu}) \frac{\partial \tilde{\nu}}{\partial x_j} \right) + C_{b2} \frac{\partial \tilde{\nu}}{\partial x_i} \frac{\partial \tilde{\nu}}{\partial x_i} \right] \,. \tag{6}$$

The turbulent eddy viscosity $\nu_t$ is directly calculated from the transported modified viscosity $\tilde{\nu}$ via:

$$\nu_t = \tilde{\nu} f_{v1} \,, \qquad f_{v1} = \frac{\chi^3}{\chi^3 + C_{v1}^3} \,, \qquad \chi = \frac{\tilde{\nu}}{\nu} \,. \tag{7}$$

The modified velocity scale $\tilde{S}$ in the production term $P_{\tilde{\nu}}$ is defined via:

$$\tilde{S} = \max \left( S_{\sigma\text{-DDES}} + \frac{\tilde{\nu}}{\kappa^2 d_w^2} f_{v2}, \ 0.3 \cdot S_{\sigma\text{-DDES}} \right) \,, \tag{8}$$

$$S_{\sigma\text{-DDES}} = S_{\text{RANS}} - f_{\text{P,ZDES}} \cdot \text{pos} \left( L_{\text{RANS}} - L_{\text{LES}} \right) \cdot \left( S_{\text{RANS}} - B_\sigma S_\sigma \right) \,, \tag{9}$$

$$S_{\text{RANS}} := \sqrt{2 \Omega_{ij} \Omega_{ij}} \,, \qquad \Omega_{ij} = \frac{1}{2} \left( \frac{\partial U_i}{x_j} - \frac{\partial U_j}{x_i} \right) \,, \tag{10}$$

$$S_\sigma := \frac{\sigma_3 \left( \sigma_1 - \sigma_2 \right) \left( \sigma_2 - \sigma_3 \right)}{\sigma_1^2} \,, \tag{11}$$

where $d_w$ is the wall distance field (i.e. local distance to the closest solid wall). In Eq. 11, $S_\sigma$ is the velocity operator of the $\sigma$ LES model of Nicoud et al. (16), which is built on the three singular values $\sigma_1 \geq \sigma_2 \geq \sigma_3 \geq 0$ of the velocity gradient tensor $g_{ij} = \partial U_i / \partial x_j$. This modification allows the model to rapidly switch from RANS to LES in separated shear layers, which is vital for an accurate flow prediction. The pos-function in Eq. 8 is defined via:

$$\text{pos}(a) = \begin{cases} 0 & \text{, if } a \leq 0 \\ 1 & \text{, if } a > 0 \end{cases} \,. \tag{12}$$

The DDES length scale used in the dissipation term $\epsilon_{\tilde{\nu}}$ is defined via:

$$L_{\text{DDES}} = L_{\text{RANS}} - f_{\text{P,ZDES}} \cdot \max \left( 0, \ L_{\text{RANS}} - L_{\text{LES}} \right) \,, \tag{13}$$

$$L_{\text{RANS}} = d_w \,, \qquad L_{\text{LES}} = \Psi C_{\text{DES}} \Delta \,, \qquad \Psi^2 = \min \left[ 10^2, \ \frac{1 - \frac{C_{b1}}{C_{w1} \kappa^2 f_w^*} f_{v2}}{f_{v1}} \right] \,, \quad (14)$$

In both the definition for the production term in Eq. 9 as well as for the dissipation term in Eq. 13, a blending function $f_{\text{P,ZDES}}$ is used which is designed to prevent the erroneous activation of the LES mode inside of attached boundary layers. This function referred to as "shielding function" is an enhanced formulation proposed by Deck & Renard as part of their ZDES approach (4) and offers a more robust behaviour relative to the original DDES shielding function $f_d$ of Spalart et al. (21). The formulation reads:

$$f_{\text{P,ZDES}} = f_d(r_d) \cdot [1 - (1 - f_d(\mathcal{G}_{\tilde{\nu}})) \cdot f_R(\mathcal{G}_\Omega)] \ , \tag{15}$$

$$f_d(x) = 1 - \tanh\left[(C_{d1}x)^{C_{d2}}\right] \ , \qquad r_d = \frac{\nu_t + \nu}{\kappa^2 d_w^2 \max\left(\sqrt{\frac{\partial U_i}{\partial x_j}\frac{\partial U_i}{\partial x_j}} \ ; \ 10^{-10}\right)} \ , \tag{16}$$

$$\mathcal{G}_{\tilde{\nu}} = \frac{C_3 \max\left(0, \ -\partial\tilde{\nu}/\partial n\right)}{\kappa d_w \sqrt{\frac{\partial U_i}{\partial x_j}\frac{\partial U_i}{\partial x_j}}}, \qquad \mathcal{G}_\Omega = \frac{\partial(|\omega|)}{\partial n}\sqrt{\frac{\tilde{\nu}}{\left(\sqrt{\frac{\partial U_i}{\partial x_j}\frac{\partial U_i}{\partial x_j}}\right)^3}} \ , \tag{17}$$

$$f_R(\mathcal{G}_\Omega) = \begin{cases} 1 & \text{, if } \mathcal{G}_\Omega \leq C_4 \\ \frac{1}{1+\exp\left(\frac{-6\alpha}{1-\alpha^2}\right)} & \text{, if } C_4 < \mathcal{G}_\Omega < \frac{4}{3}C_4 \\ 0 & \text{, if } \mathcal{G}_\Omega \geq \frac{4}{3}C_4 \end{cases} \tag{18}$$

$$\alpha = \frac{\frac{7}{6}C_4 - \mathcal{G}_\Omega}{\frac{1}{6}C_4}, \qquad C_3 = 25, \qquad C_4 = 0.03 \ . \tag{19}$$

The LES filter width $\Delta$ in the LES length scale definition of Eq. 14 is based on a formulation originally proposed by Spalart (13), which was subsequently adapted by the authors to allow for an easier implementation into the cell-centred OpenFOAM code:

$$\Delta = \tilde{\Delta}_\omega = \alpha_\Delta \cdot \max_{i=1,n}\left|2\vec{n}_\omega \times \left(\vec{f_i} - \vec{c}\right)\right| \ , \qquad \alpha_\Delta = 1.035 \ , \tag{20}$$

where $n$ is the number of faces of each cell, $\vec{n}_\omega$ is a normal vector pointing in the direction of the vorticity vector, $\vec{f_i}$ is the face centre vector of face $i$ and $\vec{c}$ is the cell centre vector. The remaining model functions read:

$$f_{v2} = 1 - \frac{\chi}{1 + \chi f_{v1}} \ , \qquad f_w = g\left(\frac{1 + C_{w3}^6}{g^6 + C_{w3}^6}\right)^{1/6} \ , \tag{21}$$

$$g = r + C_{w2}\left(r^6 + r\right) \ , \qquad r = \frac{\tilde{\nu}}{\tilde{S}\kappa^2 d_w^2} \ . \tag{22}$$

Finally, the model coefficients read:

$$C_{b1} = 0.1355 \ , \qquad C_{b2} = 0.622 \ , \qquad C_{w1} = \frac{C_{b1}}{\kappa^2} + \frac{1 + C_{b2}}{\sigma_{\tilde{\nu}}} \ ,$$

$$C_{w2} = 0.3 \ , \qquad C_{w3} = 2 \ , \qquad C_{v1} = 7.1 \ , \qquad \kappa = 0.41 \ , \qquad \sigma_{\tilde{\nu}} = 2/3 \ ,$$

$$C_{\text{DES}} = 0.65 \ , \qquad C_{d1} = 10 \ , \qquad C_{d2} = 3 \ , \qquad B_\sigma = 67.7 \ . \tag{23}$$

A more detailed discussion of the model and its components can be found in Mockett et al. (13), Fuchs et al. (7), Shur et al. (19), Nicoud et al. (16) and Deck & Renard (4).

## A.2 FLOW SOLVER AND DISCRETISATION

The flow solver employed in the scale-resolving simulations (SRS) is a derivative of the `pimpleFoam` solver of the central OpenFOAM v2212 release version, which was customised by Upstream CFD. It is based on the transient SIMPLE algorithm (17) which performs multiple sub-iterations per time step to achieve a sufficient convergence of the coupled non-linear equation system that arises from the discretised Navier-Stokes equations. The custom solver features an enhanced version (12; 15) of the original Rhie-Chow formulation (18) which offers a consistent pressure-velocity coupling and lower numerical dissipation in the LES regions. The hybrid blending scheme of Travin et al. (23) is used to discretise the convection term in the momentum equations, where a second-order central differencing scheme was employed in regions of resolved turbulence and a second-order upwind-biased scheme elsewhere. The latter scheme was also used to discretise the convection term in the $\tilde{\nu}$ transport equation of the turbulence model. Time integration was performed via a second-order accurate implicit Euler scheme. For the boundary conditions of the turbulence model at solid walls, a formulation based on Spalding's law of the wall (22) is used that is valid for arbitrary values of the non-dimensional near-wall spacing $y^+$.

## A.3 TIME-AVERAGING PROCEDURE

Due to the fact that the Navier-Stokes equations are solved in time, only snapshots of the flow are computed at each physical time step, which can vary greatly due to the chaotic nature of turbulence. To obtain meaningful flow statistics that can be further analysed for engineering purposes, averaging over multiple time steps has to be conducted. Assuming that a simulation consists of $N$ snapshots / calculated time steps, the first and second order flow statistics are then computed via:

$$\overline{\phi} = \frac{1}{N-n+1}\left\{\sum_{t=n}^{N}\phi\right\}, \qquad \overline{\phi'^2} = \frac{1}{N-n+1}\left\{\sum_{t=n}^{N}\phi^2\right\} - \overline{\phi}^2, \tag{24}$$

$$\overline{\Phi_i} = \frac{1}{N-n+1}\left\{\sum_{t=n}^{N}\Phi_i\right\}, \qquad \overline{\Phi_i'\Phi_j'} = \frac{1}{N-n+1}\left\{\sum_{t=n}^{N}\Phi_i\Phi_j\right\} - \overline{\Phi_i\Phi_j}, \tag{25}$$

where $\phi$ is an arbitrary scalar quantity (e.g. pressure) and $\Phi$ is a vector quantity (e.g. velocity). The overbar hereby denotes time-averaged values. In general, using all time steps $N$ of a simulation for computing the flow statisics is not possible, since a time interval at the start of the simulation exists in which the flow field snapshots are still affected by the non-physical initial conditions. This simulation phase is often referred to as "initial transient". The time-averaging procedure is hence only started at a certain point in time $t = n$, which has to be determined by the user based on suitable criteria. A second issue concerns the total length of the simulation after the initial transient has been bridged and time-averaging has been started, which determines both the computational expense of the simulation as well as the statistical accuracy of the flow statistics.

In this work, the statistical analysis tool Meancalc has been used for these tasks, which is developed and maintained by Upstream CFD. The tool automatically controls the runtime of each transient simulation by simultaneously evaluating the expected initial transient interval and the statistical accuracy of user-selected variables. An example plot of such an analysis is shown in Fig. 8, where the time series of the drag coefficient $C_\text{d}$ (see Sect. B) from one of the validation simulations (see Sect. C) is evaluated.

These templated Meancalc plots are also provided as part of the data set archive for all simulated cases (see Sect. D.6). In the plots, the time axis (i.e. x-axis) is scaled both in physical simulation time (i.e. ~seconds) as well as in non-dimensional units referred to as "convective time units" (CTU). The normalisation is achieved by multiplying the physical simulation time $t$ with a characteristic velocity and time scale, i.e. CTU $= t \times U_0/l_0$. For this application, the characteristic velocity scale is the freestream velocity $U_0 = U_\infty$ and the characteristic length scale is set to be the wheelbase of the car $l_0 = l_\text{wheelbase}$ (see Fig. 10). This enables a better comparability between different car geometries, since the time a fluid particle needs to completely pass over the body surface is usually a good measure for the physical scales of the flow.

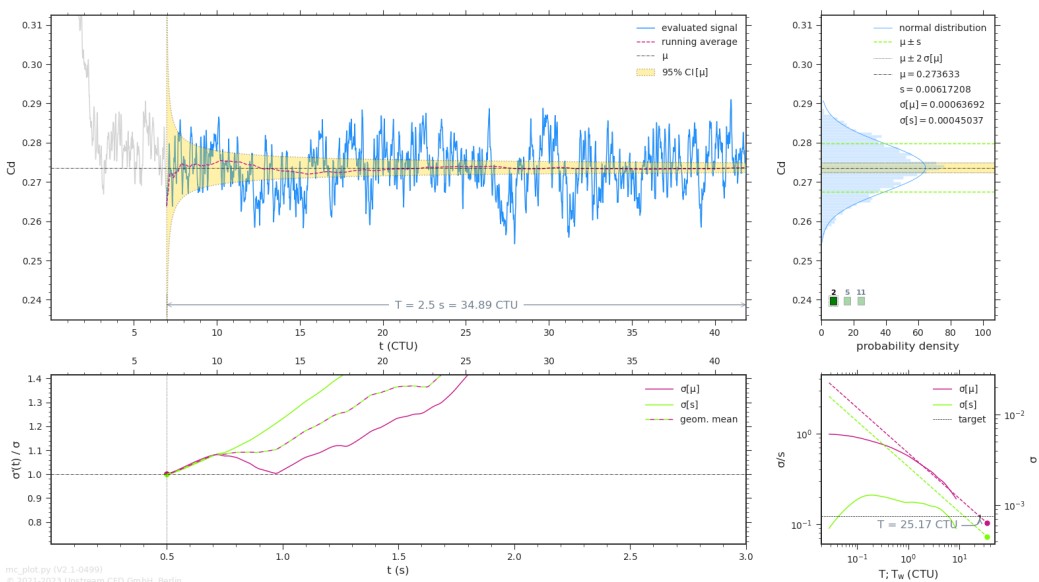

Figure 8: Example plot for evaluation of the time series of the drag coefficient $C_\mathrm{d}$ via the statistical analysis tool Meancalc. The time series originates from the validation simulation of the DrivAer 2a case.

In the upper left hand plot of Fig. 8, the analysed time series is plotted. The time interval that has been determined as statistically steady-state and selected for time-averaging by the algorithm is coloured in blue, and the discarded initial part of the signal is greyed out. The calculated time-averaged mean value $\mu$ is plotted via a dashed black line. In addition, the running average is plotted via the dashed light-purple line. The Meancalc tool also computes a 95% statistical confidence interval (CI) based on the analysed time series, and the shrinking of the CI with growing sample size is visualised via the yellow-coloured area in Fig. 8.

The final values of the mean $\mu$, its standard deviation $s$ and the confidence intervals $\sigma[\mu]$ and $\sigma[s]$ are provided in the upper right-hand side plot in Fig. 8, where 95% CI$[\mu] = 2\sigma[\mu]$. This plot also visualises the binned distribution of the signal around the final mean value as blue bars overlayed with the theoretical normal distribution as blue line, the final mean value as dashed purple line, its standard deviation as dashed green line and the 95% CI again as yellow-coloured area. The small boxes in the lower left corner are trend indicators of the signals evaluated in the transient detection. Green indicates a lower probability of a trend while orange or red indicate a higher probability. If a box is red, the corresponding signal does not fulfil the assumptions for the calculation of the confidence intervals.

The lower two plots in Fig. 8 give additional insight into the criteria for detecting initial transient (lower left) and information on the reduction of error with increasing sample size (lower right). The lower left diagram shows the evolution of the relative estimated error $\sigma(t)/\sigma$ for the mean value, its standard deviation and the geometric mean of both normalised by the respective final estimated error. The lower right plot shows when the target accuracy has been reached, or if not, when it is estimated to be reached. More background on the underlying mathematics is given in ref. (14).

The following criteria have been applied for the automatic runtime control of each simulation:

1. **Initial transient**: Time series of the three non-dimensional coefficients for drag ($C_\mathrm{d}$), lift ($C_\mathrm{l}$) and side force ($C_\mathrm{s}$) are analysed to determine the overall initial transient. The initial transient portion is then truncated from the overall flow field average using periodic checkpoints. A minimum time sample of at least 25 CTUs is used in the initial transient detection to avoid a premature termination of the simulation due to spurious effects of short time series.

2. **Target statistical accuracy**: The statistical accuracy of the drag coefficient is used as a criterion, where the simulation is stopped when a target accuracy (i.e. 95% confidence interval) of $\pm 1.5$ drag counts has been reached in the transient-free (i.e. statistically steady-state) portion of the simulation. An automotive drag count corresponds to a change in drag coefficient of $\Delta C_{\mathrm{d}} = \pm 0.001$. In addition, the overall simulation length has been limited to $40 \leq t \times U_\infty / L \leq 60$ to prevent excessive hardware usage for outlier cases with insufficient statistical convergence. However less than 5% of the cases reached this limit. Figure 9 shows an example of the statistical history from runs 1 to 18, to give a larger sense of the typical averaging times.

This procedure ensures that the statistical accuracy of all simulations is comparable. Adapting the simulation length dynamically to obtain a desired target accuracy is a novel approach of particular value, as we expect that statistical consistency is important in a machine learning training dataset. In case it becomes evident that a lower error threshold is required, the simulations can also be more easily re-run with this mechanism in place by simply decreasing statistical error target.

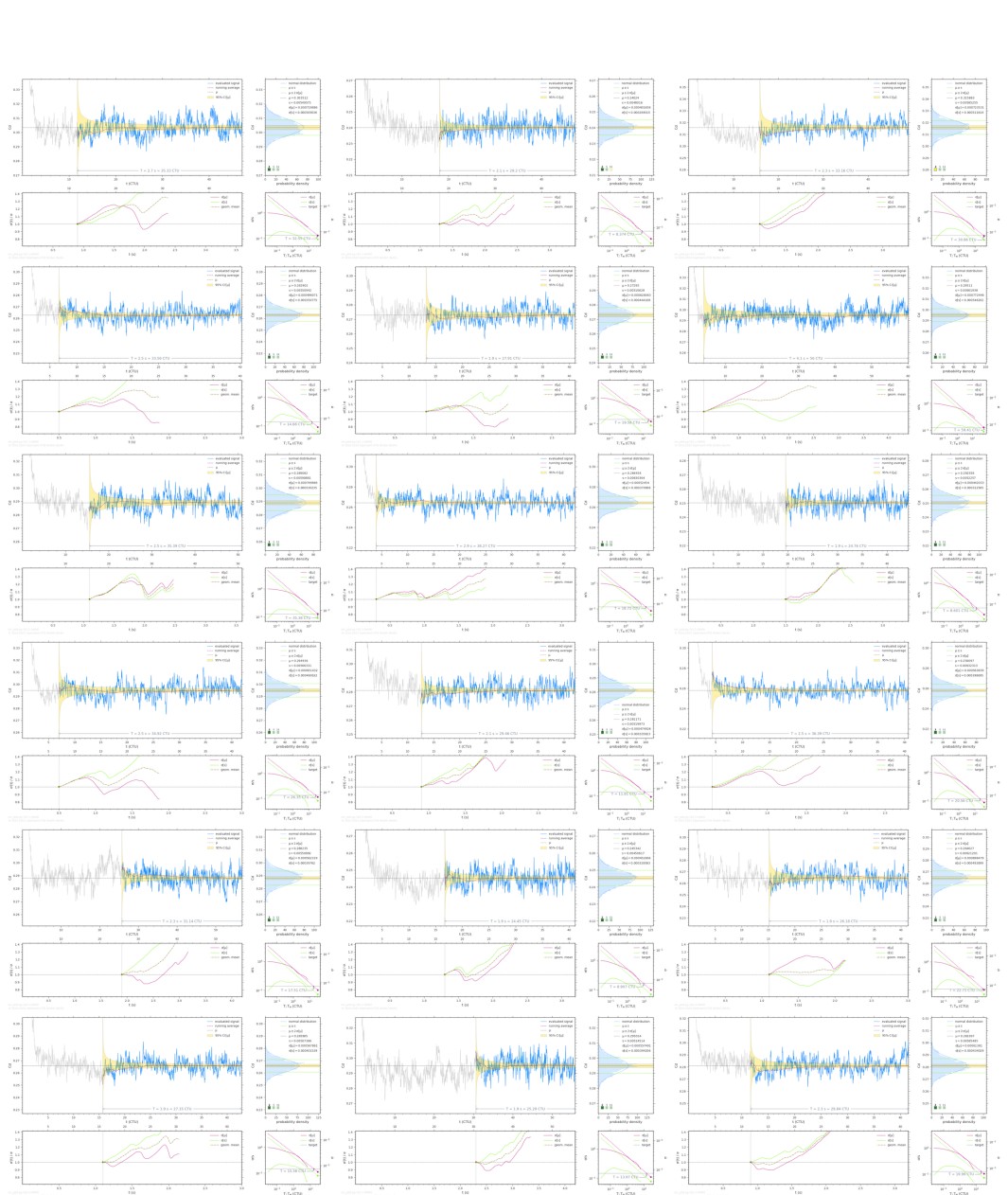

Figure 9: Initial transient detection and time-averaging periods shown for drag coefficient traces for runs 1 to 18

## B    DEFINITION OF AERODYNAMIC QUANTITIES USED IN THE DATASET

### B.1    FORCE AND MOMENT COEFFICIENTS

In fluid dynamics, dimensionless quantities are often used to allow for a comparison between different geometries and flow conditions in a more systematic manner. For automotive aerodynamics, the integral forces and moments acting on the car are of prime interest. In Fig. 10, the nomenclature and coordinate system used in this work is introduced.

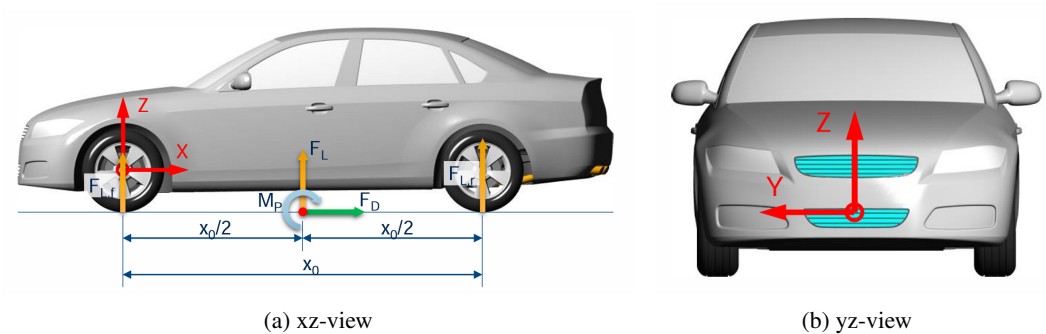

(a) xz-view                                   (b) yz-view

Figure 10: Definition of aerodynamic forces in CFD coordinate system used in this work.

The total forces $F_{(x,y,z)}$ and moments $M_{(x,y,z)}$ are hereby typically given as coefficients in non-dimensional form, where the local forces / moments are first integrated over the entire surface $S$ of the vehicle:

$$F_{(x,y,z)} = \int_S F_i^{\mathrm{tot}} \, n_{(x,y,z),i} \, dS \, , \tag{26}$$

$$M_{(x,y,z)} = \int_S \left( e_{ijk} \left( x_j - x_{\mathrm{ref},j} \right) F_k^{\mathrm{tot}} \right) n_{(x,y,z),i} dS \, , \tag{27}$$

where $x_{\mathrm{ref},i}$ is the reference point for the calculation of the moments, $n_{(x,y,z),i}$ are the unit vectors pointing in the $x$, $y$ and $z$ directions (i.e. $n_x = [1,0,0]$, $n_y = [0,1,0]$, $n_z = [0,0,1]$), $x_i$ is the surface coordinates vector and $e_{ijk}$ is the Levi-Civita symbol. The aerodynamic forces on the body are composed of two contributions, a normal pressure force $F_i^p$ and a viscous force $F_i^v$ resulting from the wall shear stresses:

$$F_i^{\mathrm{tot}} = F_i^p + F_i^v \, , \tag{28}$$

$$F_i^p = (p - p_{\mathrm{ref}}) \, S_{n,i} \, , \quad F_i^v = \tau_{w,i} = \rho_\infty \left( \nu + \nu_t \right) \left[ \frac{\partial U_i}{\partial x_j} + \frac{\partial U_j}{\partial x_i} \right] S_{n,j} \, , \tag{29}$$

where $S_{n,i}$ is the surface normal vector and $\tau_{w,i}$ is the wall shear stress force vector. The normalised force coefficients are then obtained by normalising the force components in the defined spatial directions with a dynamic reference pressure $p_{\mathrm{dyn,ref}}$, a reference area $A_{\mathrm{ref}}$ and a reference length scale $L_{\mathrm{ref}}$ (see Section D.6 for details on the two different files (force_mom_i.csv and force_mom_constref_i.csv) that are provided in the dataset based upon constant $A_{ref}$ & $L_{ref}$ or one based upon each geometry)

$$C_{\mathrm{d}} = \frac{F_x}{p_{\mathrm{dyn,ref}} A_{\mathrm{ref}}} \;, \qquad C_{\mathrm{l}} = \frac{F_z}{p_{\mathrm{dyn,ref}} A_{\mathrm{ref}}} \;, \qquad C_{\mathrm{s}} = \frac{F_y}{p_{\mathrm{dyn,ref}} A_{\mathrm{ref}}} \;, \tag{30}$$

$$C_{\mathrm{m,roll}} = \frac{M_x}{p_{\mathrm{dyn,ref}} A_{\mathrm{ref}} l_{\mathrm{ref}}} \;, \qquad C_{\mathrm{m,pitch}} = \frac{M_z}{p_{\mathrm{dyn,ref}} A_{\mathrm{ref}} l_{\mathrm{ref}}} \;, \tag{31}$$

$$C_{\mathrm{m,yaw}} = \frac{M_y}{p_{\mathrm{dyn,ref}} A_{\mathrm{ref}} l_{\mathrm{ref}}} \;, \qquad p_{\mathrm{dyn,ref}} = \frac{1}{2} \rho_\infty |U_\infty|^2 \;, \tag{32}$$

The primary force coefficients are hence given by the drag coefficient $C_{\mathrm{d}}$, the lift coefficient $C_{\mathrm{l}}$ and the side force coefficient $C_{\mathrm{s}}$. The three moment coefficient are referred to as roll moment coefficient $C_{\mathrm{m,roll}}$, pitch moment coefficient $C_{\mathrm{m,pitch}}$ and yaw moment coefficient $C_{\mathrm{m,yaw}}$. For assessing the aerodynamic characteristics of a car, the total lift coefficient is often additionally split into two parts which represent the respective contributions of the lift acting on the front and rear axles:

$$C_{\mathrm{lf}} = \frac{1}{2} C_{\mathrm{l}} + \frac{C_{\mathrm{m,pitch}}}{l_{\mathrm{ref}}} \;, \quad C_{\mathrm{lr}} = \frac{1}{2} C_{\mathrm{l}} - \frac{C_{\mathrm{m,pitch}}}{l_{\mathrm{ref}}} \;, \tag{33}$$

where $C_{\mathrm{lf}}$ is the front lift coefficient and $C_{\mathrm{lr}}$ is the rear lift coefficient. It is important to note that the values of all presented coefficients depend on the reference parameters $A_{\mathrm{ref}}$, $l_{\mathrm{ref}}$ and $\vec{x}_{\mathrm{ref}}$.

### B.2 PRESSURE AND SKIN-FRICTION

Equivalent to the integral force coefficients, normalisation is often also applied to specific flow fields to account for scale effects between different configurations. A frequently analysed quantity is the dimensionless static pressure coefficient $C_p$, which describes the relative static pressures throughout the flow field normalised by the dynamic reference pressure. It is defined as:

$$C_p = \frac{p - p_{\mathrm{ref}}}{0.5 \rho_\infty |U_\infty|^2} \;. \tag{34}$$

Example plots of $C_p$ for the volume and surface fields are given in Figs. 24b and 24f respectively. Note that the value of $C_p$ inherently depends on the reference pressure value $p_{\mathrm{ref}}$, which is often extracted at a specific point of the domain (typically in a freestream region away from the wall). In our CFD, the pressure reference point is located at $x_{\mathrm{PR}} = 80\,\mathrm{m}$, $y_{\mathrm{PR}} = z_{\mathrm{PR}} = 10\,\mathrm{m}$ on the outlet boundary patch, where $p_{\mathrm{ref}} = 0\,\mathrm{Pa}$. Equivalent to the static pressure coefficient, the total pressure coefficient $C_{pt}$ can be defined as follows:

$$C_{pt} = \frac{p_t - p_{\mathrm{ref}}}{0.5 \rho_\infty |U_\infty|^2} \;, \quad p_t = p + 0.5 \rho_\infty |U_i|^2 \;, \tag{35}$$

where the given definition for the total pressure $p_t$ is valid for an incompressible fluid. Likewise, the wall friction coefficient $C_f$ is a frequently used quantity, which represents the normalised magnitude of the wall shear stress vector:

$$C_f = \frac{|\tau_{w,i}|}{0.5 \rho_\infty |U_\infty|^2} \;. \tag{36}$$

See Fig. 24h for an example visualisation of $C_f$. A dimensionless quantity specifically used in automotive aerodynamics is the micro drag or local drag coefficient $C_{\mathrm{dl}}$ first defined by Cogotti (3). This quantity is designed to identify areas in wake regions which contribute most to the overall drag (i.e. areas with $C_{\mathrm{dl}} > 0.5$). It is defined via the total pressure coefficient and the velocity vector:

$$C_{\mathrm{dl}} = 1 - C_{pt} - \left(1 - \frac{U_x}{|U_\infty|}\right)^2 + \frac{U_y^2 + U_z^2}{|U_\infty|^2} \;. \tag{37}$$

where $x$ is the streamwise direction and $y/z$ are the two perpendicular directions. An example is shown in Fig. 24d.

## C    EXTENDED VALIDATION RESULTS OF NUMERICAL METHODOLOGY

This section complements the validation results presented in Section 3.3 of the main paper with an extended analysis. Validation of the CFD workflow is carried out for the Ford OCDA DrivAer model (10), which is one of the two test cases of the 4th Automotive CFD Prediction Workshop ("AutoCFD-4")[7]. In the following, we demonstrate that the CFD methodology presented in the paper provides a reliable correlation to high-quality experimental data, which builds confidence that the approach can also reasonably predict the 500 other geometric variations of the DrivAer (see Sect. D.7).

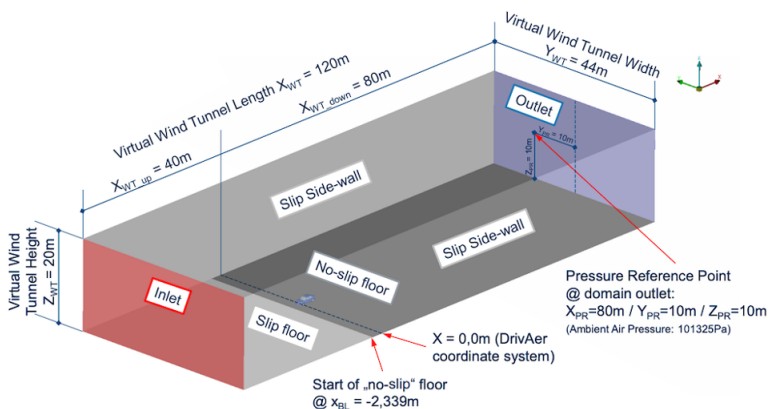

Figure 11: Sketch of computational domain used in the CFD setup (11).

Since its original publication by Heft et al. (8), numerous wind-tunnel campaigns have been conducted for the DrivAer model. In this work, we validate against the experimental data of Hupertz et al. (9), where an extensive wind tunnel campaign was conducted in the Pininfarina facility. In CFD, the computational domain and boundary conditions are often adjusted to either reproduce open road or experimental conditions. In this case, a large open-road domain, sketched in Fig. 11 has been chosen, which is representative of the types of setups commonly used in the automotive industry. To mimic the ground boundary layer present in the Pininfarina wind tunnel, the domain floor has been divided into two parts, an inviscid part upstream of the car ("Slip floor") and a viscous wall starting at $x_{\mathrm{BL}} = -2.339$ m ("No-slip floor"), which is located 2.346 m upstream of the front axle. However, geometrical details of the wind tunnel, such as the upstream nozzle (with $11\,\mathrm{m}^2$ area) and downstream collector of the finite open test section, are not integrated into the CFD setup. The side and top patches of the CFD domain are prescribed as inviscid walls, and blockage ratios can be considered negligible ($\approx 0.25\%$) thanks to the large domain. This computational domain has been used for all dataset simulations.

An important further aspect of the CFD setup concerns the location of the reference pressure $p_{\mathrm{ref}}$, which is set at the outlet patch that located 80 m downstream of the start of the viscous wall patch (see Fig. 11). Due to the friction losses associated with the car wake and the ground boundary layer, the static pressure continuously decreases in the downstream direction, so that shifting the reference pressure location in the x-direction would result in different values of $p_{\mathrm{ref}}$. It is important to remember when processing the provided data that the absolute local kinematic pressure values $p^* = p/\rho_\infty$ contained in the dataset are influenced by the specific boundary conditions used in this setup. This also concerns other quantities derived from the $p^*$ field such as the static and total pressure coefficients $C_p$ and $C_{pt}$. This means that e.g. surface distributions of $C_p$ provided in the database might not be strictly comparable to other cases in which different boundary conditions were used. In contrast, this issue is less relevant for the integral force coefficients, for which only the relative pressure distribution on the surface is relevant (a constant offset in $p_{\mathrm{ref}}$ is cancelled out in the integration).

---

[7]https://autocfd.org/

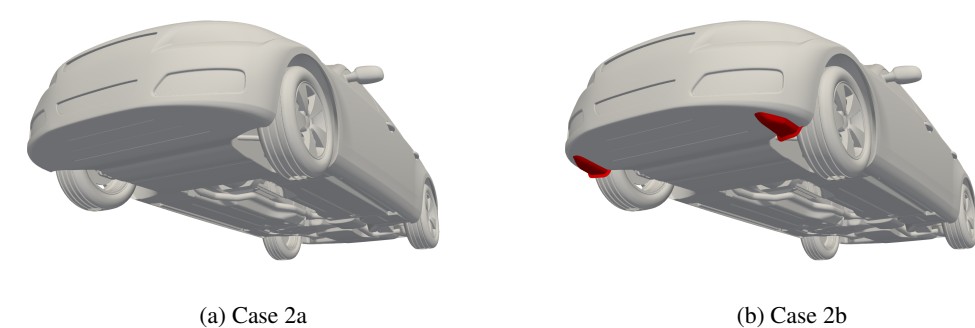

(a) Case 2a                                  (b) Case 2b

Figure 12: Visualisation of configurations 2a and 2b of Ford OCDE DrivAer model.

One of the central aims of CFD and numerical tools in general is to accurately predict deltas between different geometry designs, which is key to introducing these tools into the certification process for road vehicles. Two different configurations of the Ford OCDA DrivAer model denoted "2a" and "2b" (corresponding to their case designation in the 3rd & 4th AutoCFD workshops) were therefore used in the validation. Variant 2b includes additional front wheel air deflectors (FWAD) relative to the baseline 2a (highlighted in red in Fig. 12).

Both configurations have been simulated with exactly the same CFD workflow as the other 500 geometries of the dataset, where the simulation runtime was controlled via the statistical analysis tool Meancalc (see Sect. A.3). For the simulation of the DrivAer 2a case, this resulted in a simulated time of $3.0\,\mathrm{s} \approx 42\,\mathrm{CTU}$, whereas the DrivAer 2b case ran for $3.6\,\mathrm{s} \approx 50\,\mathrm{CTU}$. In Fig. 13, the automatically generated Meancalc plots of the drag coefficient time series analysis are shown. It can be seen that the initial transient has been successfully removed from both evaluated signals. Furthermore, the two simulations have been automatically stopped after the target accuracy of $2\sigma(C_\mathrm{d}) = 95\%\,\mathrm{CI}(C_\mathrm{d}) \le 0.0015$ was reached.

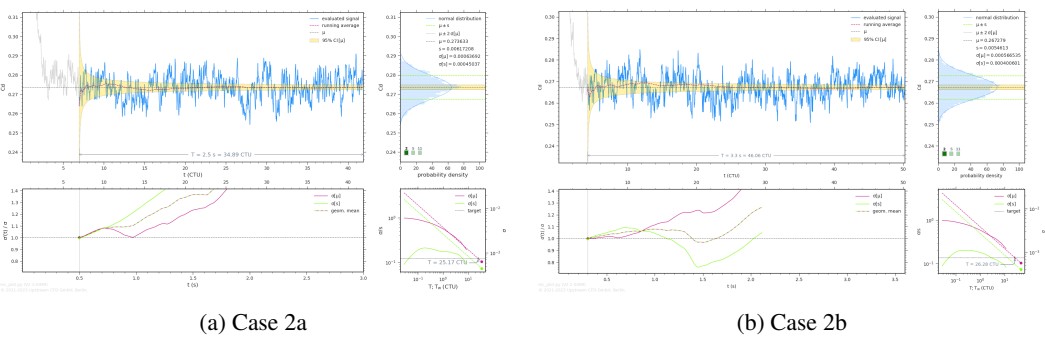

(a) Case 2a                                  (b) Case 2b

Figure 13: Meancalc generated plots showing the statistical evaluation of the time series of the drag coefficient $C_\mathrm{d}$ for the two validation simulations of the Ford OCDA DrivAer.

One of the primary quantities of interest in car design are the integral force coefficients. Table 1, compares different force coefficients between experiment and CFD, including the effect of the FWAD (case 2b relative to 2a). For additional perspective, we also refer to third-party CFD results from the 3rd AutoCFD workshop for the same test cases, which were computed by numerous (anonymised) participants and codes. This valuable resource is freely available via an interactive online dashboard[8], from which we show the absolute force coefficients for case 2a (Fig. 14) and the deltas for case 2b relative to case 2a (Fig. 15). The red data points highlight results from SRS methods on the committee mesh from that workshop, which is an earlier version of the current mesh giving very similar results.

Table 1: Comparison of time-averaged forces coefficients between Ford experiment (9) and simulation approach of the CFD workflow. $\Delta$(2b-2a) denotes the effect of the FWAD (change in the case 2b value relative to the 2a value).

| | Part | | | |
| Case | CFD/Exp | $C_{\mathrm{d}}$ | $C_{\mathrm{l}}$ | $C_{\mathrm{lf}}$ | $C_{\mathrm{lr}}$ |
| --- | --- | --- | --- | --- | --- |
| 2a | Exp: Hupertz et al. (2021) | 0.255 | 0.087 | -0.023 | 0.111 |
| 2b | Exp: Hupertz et al. (2021) | 0.242 | 0.082 | -0.019 | 0.101 |
| $\Delta$(2b-2a) | Exp: Hupertz et al. (2021) | -0.013 | -0.005 | 0.004 | -0.010 |
| 2a | CFD: SRS | 0.274 | 0.033 | -0.073 | 0.106 |
| 2b | CFD: SRS | 0.267 | 0.039 | -0.064 | 0.103 |
| $\Delta$(2b-2a) | CFD: SRS | -0.007 | 0.006 | 0.009 | -0.003 |

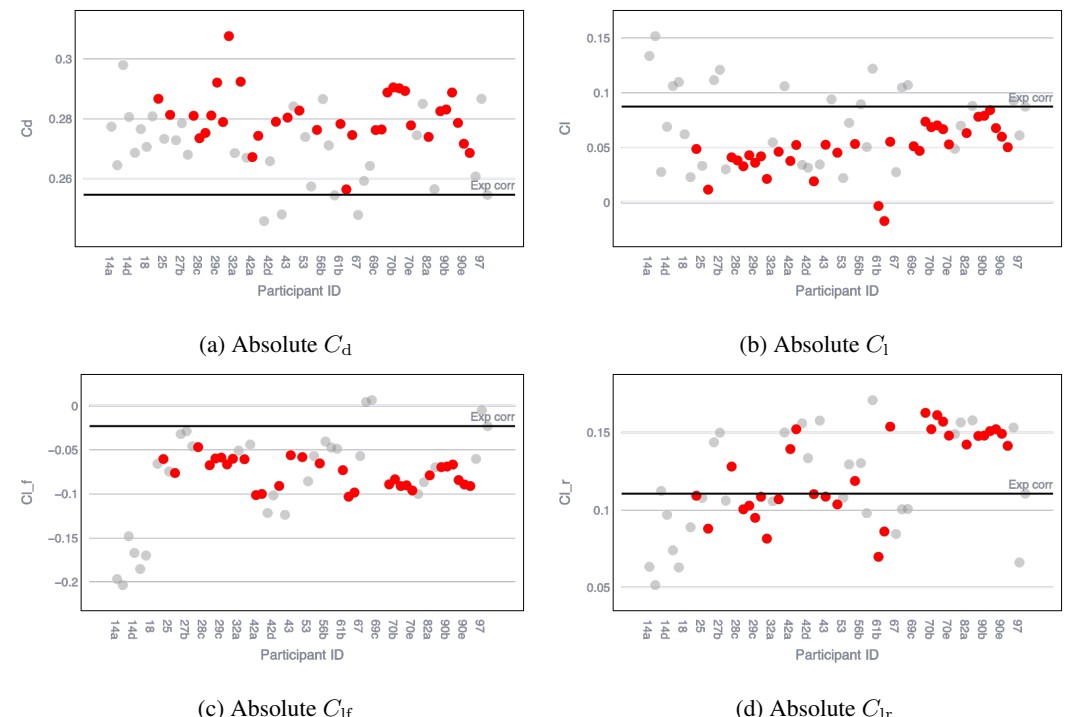

(a) Absolute $C_{\mathrm{d}}$      (b) Absolute $C_{\mathrm{l}}$

(c) Absolute $C_{\mathrm{lf}}$      (d) Absolute $C_{\mathrm{lr}}$

Figure 14: Absolute force coefficients for all CFD results from the 3rd AutoCFD workshopfn:AutoCFDDashboard (symbols) compared to the Ford experiment (9) (black line), ordered by anonymous participant ID. Results from SRS methods on the committee ANSA mesh are highlighted (red symbols).

---

[8]https://auto-cfd-workshop-3.cfdsolutions.net/

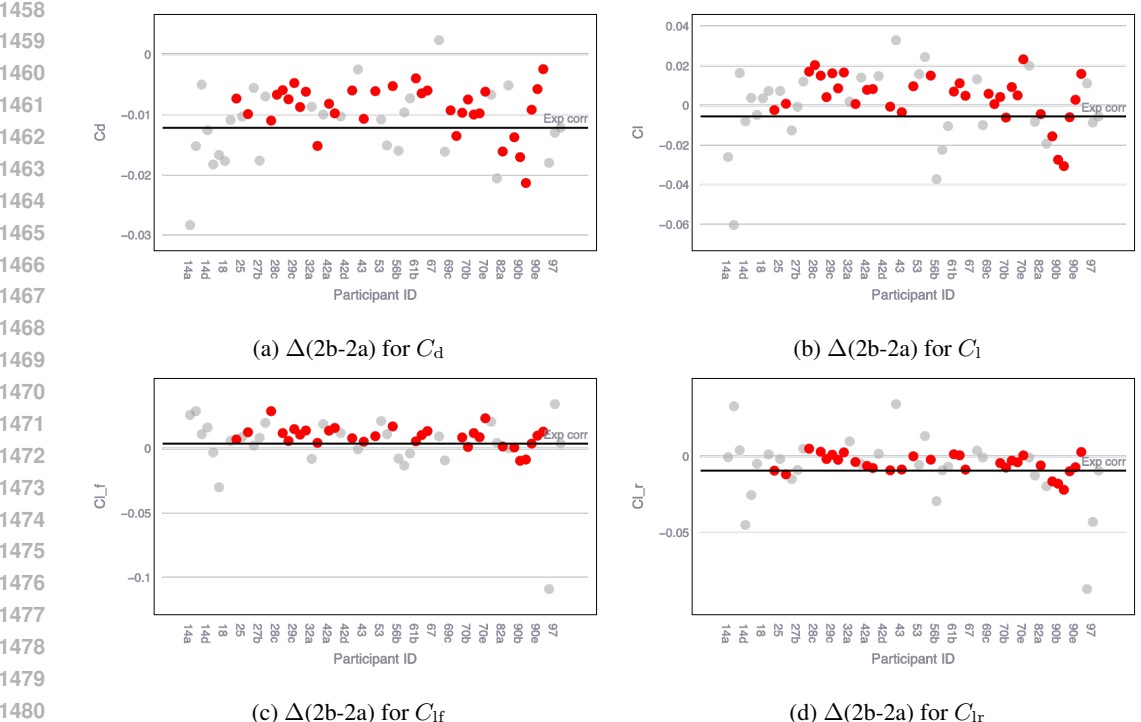

(a) $\Delta$(2b-2a) for $C_d$

(b) $\Delta$(2b-2a) for $C_l$

(c) $\Delta$(2b-2a) for $C_{lf}$

(d) $\Delta$(2b-2a) for $C_{lr}$

Figure 15: Deltas of force coefficients for case 2b relative to case 2a (effect of FWAD) for all CFD results from the 3rd AutoCFD workshopfn:AutoCFDDashboard (symbols) compared to the Ford experiment (9) (black line), ordered by anonymous participant ID. Results from SRS methods on the committee ANSA mesh are highlighted (red symbols).

The drag coefficient is relevant for the $CO_2$ emissions and hence certification. The predicted drag coefficient is higher than experiment by roughly 20 counts[9] in both simulations, and the FWAD effect is weaker than in the experiment (Tab. 1). All SRS results from the 3rd AutoCFD workshop likewise predict higher drag than the experiment (Fig. 14a), and most (82%) also predict a weaker FWAD effect (Fig. 15a). The acceptance criterion for CFD defined by the United Nations regulation ECE 154, also known as WLTP[10], is an accuracy threshold of $\delta_{WLTP} := |\Delta(C_dA)_{CFD} - \Delta(C_dA)_{EXP}| \le 0.015 \text{ m}^2$ for the prediction of deltas between two designs (where $A$ is the car frontal area). This criterion is fulfilled by the CFD approach, which returns a certification value of $\delta_{WLTP} = 0.013 \text{ m}^2$ for the delta between configurations 2a and 2b. It should be noted, however, that car manufacturers strive for a much higher accuracy of 1-2 drag counts in predictions of $\Delta(C_d)$, since every single count directly impacts the energy efficiency of a car.

No specific criterion is targeted for the lift prediction, but $C_l$ is around 50 counts lower in the CFD (Tab. 1). This deviation mainly originates in the prediction of front lift (i.e. $C_{lf}$), whereas the rear lift is in close agreement. Most SRS contributions to the 3rd AutoCFD workshop show the same discrepancies to similar levels. This may be related to uncertainties in the comparability between CFD and experiment: "pad corrections" to the lift measurement, to account for the aerodynamic force acting on the wheel belt surface near the tyre contact patch, are a topic of ongoing research (25).The front lift is a very sensitive quantity, with a spread of 48 counts (spanning positive and negative values) observed between three different wind tunnels by Hupertz et al. (11).

When looking at the delta prediction of the lift, it stands out that the change for case 2b relative to case 2a is positive in the simulation while it is negative in the experiment. This is also the case for 73% of the AutoCFD SRS results (Fig. 15b). The CFD deltas of front and rear lift individually have

---

[9]An automotive "count" corresponds to a change in force coefficient of 0.001

[10]WLTP = Worldwide harmonized Light vehicles Test Procedure, `https://unece.org/transport/vehicle-regulations-wp29/standards/addenda-1958-agreement-regulations-141-160`

the same sign as the experiment. In the simulations, the FWAD seems to have a stronger impact on the front lift than on the rear lift, which seems to be the opposite in the experiment.

While the prediction of integral forces is an important metric for the assessment of a simulation approach, it can also be misleading due to the potential for error cancellation. A good agreement of the force coefficients with experimental benchmark data does not necessarily mean an overall good prediction of the flow field. It is also hard to understand the causes of differences in the integral forces without further analysis. It is therefore important to also compare local quantities.

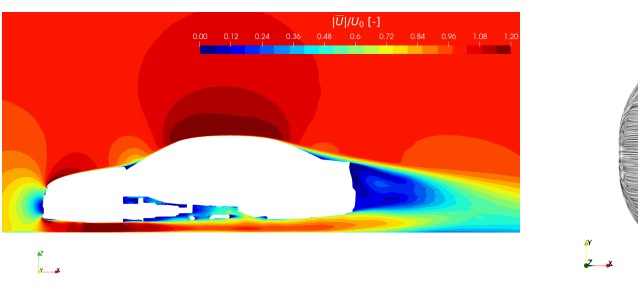
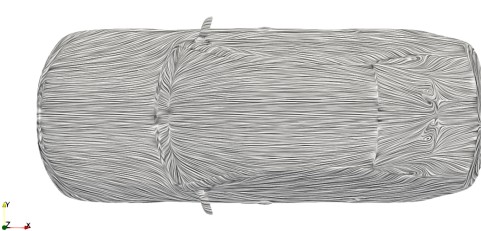

(a) Case 2a: $|\overline{U}|/U_\infty$ on $y = 0\,\mathrm{m}$ slice

(b) Case 2a: Surface streamlines

Figure 16: Contour plots for an general impression of the flow field of the DrivAer case 2a. (a) Time-averaged normalised velocity on the center plane. (b) Top view of time-averaged surface streamlines.

Fig. 16 gives an impression of the time-averaged flow field for case 2a, where contour plots of the normalised velocity magnitude on the centre plane and a top view showing surface streamlines are depicted. Equivalent plots for case 2b look almost identical since the influence of the FWAD on the flow is minimal here. The velocity contour plot shows the flow deceleration at the front of the vehicle near the stagnation point and subsequent acceleration over and under the vehicle. Underneath the car, a small recirculation region can be seen at the leading edge, which has a significant influence on the front lift. On the upper side, one of the most challenging regions to predict is the notch at the base of the rear window, where the incoming boundary layer from the roof is subjected to an adverse pressure gradient and thickens considerably. The surface streamlines in this region indicate flow separation and reattachment at the notch, which is also indicated by a pressure plateau in the experiment (Fig. 17a, discussed later). The CFD flow topology shows a symmetric pair of recirculation regions either side of the $y = 0$ centreline, where the flow is nominally attached.

In Fig. 17, a comparison between CFD and experiment is shown for the time-averaged static pressure coefficient distributions on the upperbody and underbody centrelines (only discrete probe positions are shown for all data). The prediction of the separation behaviour on the rear window (i.e. $2.7\,\mathrm{m} < x < 3.3\,\mathrm{m}$ on the upperbody) is especially challenging for CFD, but the simulation approach adopted in this work manages to reasonably predict this. Looking closely, it can be seen that the experimental pressure distribution features a small plateau in the separation region on the rear window ($3.0\,\mathrm{m} < x < 3.5\,\mathrm{m}$) which is less pronounced in the simulations due to the nominally attached centreline flow revealed in Fig. 16b and described earlier. Despite small differences in the separation region, the overall agreement for the pressure probes on the upperbody centreline is encouraging. Compared to the experiment, a mild shift of the pressure level on the roof (i.e. $1.8\,\mathrm{m} < x < 2.6\,\mathrm{m}$) can be observed. This is at least partially due to a slight systematic negative offset in the pressure coefficient data between CFD and experiment, which most likely originates from inconsistency in the reference pressure locations between CFD and the experiment (at the outlet plane in CFD; in the test section plenum, outside the flow stream, adjacent to and upstream of the nozzle in the experiment). The small deltas between cases 2a and 2b for the probe positions plotted in Fig. 17a highlight that the effect of the FWAD is negligible on the upper side of the vehicle.

The underbody centreline, plotted in Fig. 17b, exhibits generally good agreement in the pressure distribution between CFD and experiment. The effect of the FWAD seems to be somewhat less

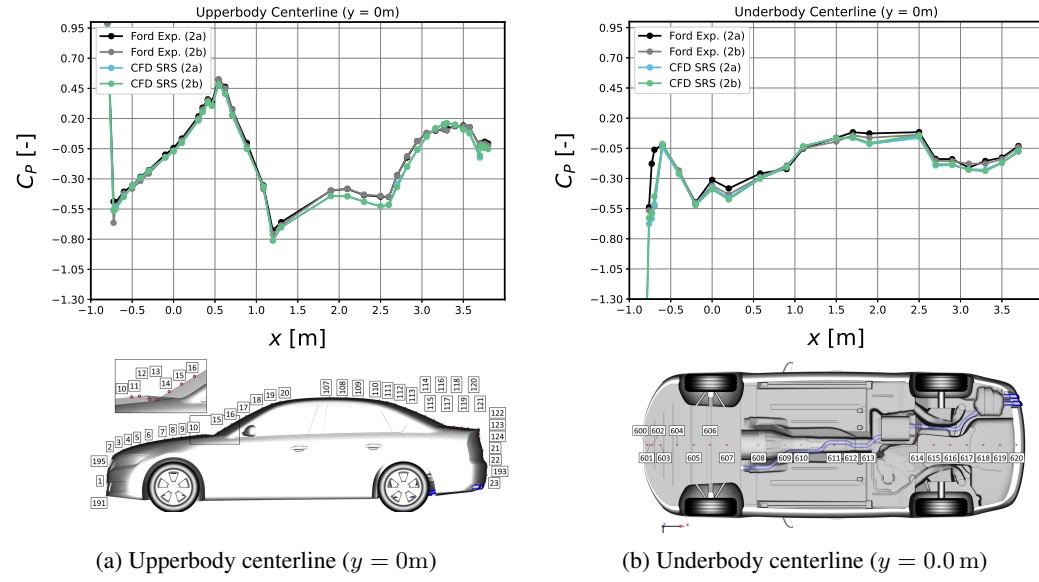

(a) Upperbody centerline ($y = 0$m)  (b) Underbody centerline ($y = 0.0$ m)

Figure 17: Comparison of time-averaged static pressure coefficient $\overline{C}_p$ for probe positions on centreline between cases 2a and 2b. Lower figures showing probe positions (red dots) reproduced with permission from (9).

pronounced in the CFD. Stronger differences can be observed at the leading edge of the underbody, where the recirculating flow region seen in the CFD is less prominent in the experiment. It is important to note that (like the front lift, as mentioned earlier) this is a very sensitive phenomenon, shown in experiments to exhibit a significant Reynolds number and wind tunnel dependency (9).

It is perhaps unsurprising that this region is also sensitive to simulation parameters. Indeed, initial tests with a two times finer time step of $\Delta t = 1 \times 10^{-4}$ s showed a closer agreement to experiment. Since this region features the some of the highest CFL numbers in the domain[11], which is a measure for the local balance between temporal and spatial resolution of a simulation, the coarser time step has a stronger effect on the prediction here. Insufficient temporal resolution can have a stabilising effect on the separated shear layer, supressing resolved turbulence and delaying reattachment to further downstream.

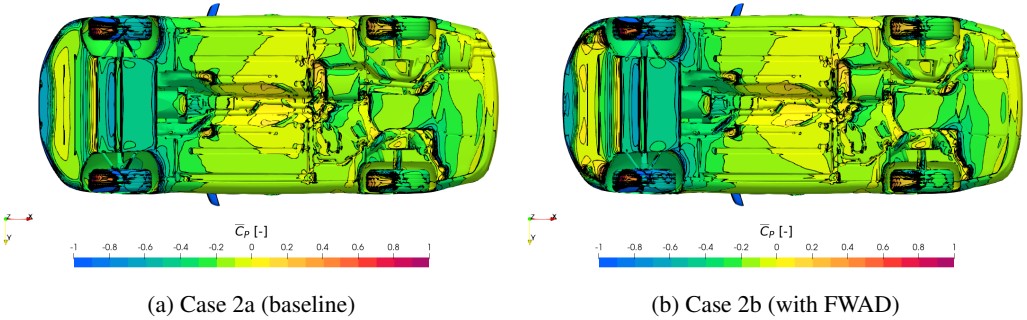

(a) Case 2a (baseline)  (b) Case 2b (with FWAD)

Figure 18: Contours of time-averaged surface pressure coefficient from CFD simulations, view from below.

The distribution of time-averaged pressure coefficient over the entire underfloor region is compared between cases 2a and 2b in Fig. 18. The FWADs have a marked non-local influence, with additional positive pressure lobes appearing just behind the leading-edge recirculation region either side of the

---

[11]Due to the fine mesh and high velocity. The Courant–Friedrichs–Lewy (CFL) number is defined as $\text{CFL} = U \cdot \Delta t / \Delta x$.

lateral centreline. The lateral strip of negative pressure on the underbody just upstream of the front wheel axle location is wider in the case with FWAD, and the mild positive pressure region near the centre of the underfloor is more extensive. These features are missed when focussing only on the centreline.

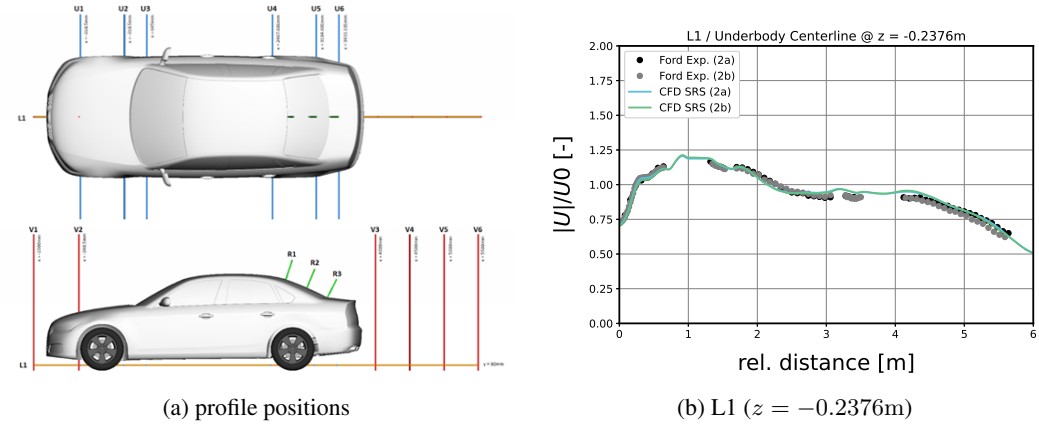

(a) profile positions                                    (b) L1 ($z = -0.2376$m)

Figure 19: (a) Positions of velocity profiles (11). (b) Comparison of the normalised velocity magnitude $|\overline{U}|/U_\infty$ on centerline velocity profile L1 under the vehicle of Case 2a and 2b.

Fig. 19b shows the comparison of the velocity magnitude distributions on the measurement profile line L1 that extends in the streamwise direction below the underfloor of the vehicle (the profile line locations are visualised in Fig. 19a). Due to the complex geometry of the underfloor as well as the complex interaction between the separated flow structures from the front region with the ground boundary layer, this area is generally challenging to accurately predict in the CFD, especially for lower fidelity turbulence modelling approaches such as steady-state RANS (11). The excellent agreement with the experiment achieved by the employed SRS approach is therefore very encouraging. However, the mild effect of the FWAD seen in the experiment in the diffusor region (4 m $< x <$ 6 m) is not apparent in the CFD.

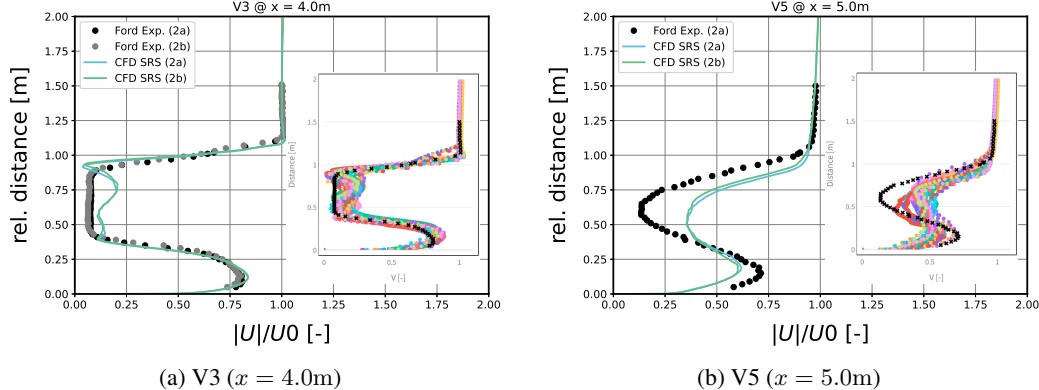

(a) V3 ($x = 4.0$m)                                      (b) V5 ($x = 5.0$m)

Figure 20: Comparison of vertical wake profiles of normalised velocity magnitude for cases 2a and 2b. Results from the 3rd AutoCFD workshopfn:AutoCFDDashboard (SRS contributions on the high-Re committee grid) are overlayed.

The momentum loss in the wake region behind the car is linked to the generated drag, so that an accurate prediction of not only surface quantities but also the volume flow field is important. In Fig. 20, two vertical wake profiles on the centreline at the experimental streamwise positions V3 and V5 are compared. At the first vertical position V3, closely behind the vehicles, the velocity gradient in the upper shear layer is slightly steeper in the simulations, perhaps reflecting a slightly smaller flow separation at the upstream notch (c.f. Fig. 17a). The velocity plateau seen in the experimental data is not apparent in the CFD, which shows undulations in the low-velocity region. Interestingly,

all CFD data from the 3rd AutoCFD workshop also show similarly undulating profiles here. Larger differences to the experiment can be seen at the downstream position V5, where both SRS show a weaker velocity deficit, the upper edge of the wake is lower, and the velocity peak emanating from the underfloor is weaker. Again, the SRS contributions from the 3rd AutoCFD workshop exhibit the same phenomena as our CFD. Although different flow solvers and turbulence models were used (including nominally higher-fidelity methods such as WMLES), the differences to the experimental measurements for the V3 and V5 positions appear to be systematic. This implies that differences between the CFD and experimental domains cause the deviations. From photographs (see e.g. (9; 11)), the wind tunnel collector appears close to the vehicle wake and can be expected to influence the flow here. For both centreline velocity profiles, the influence of the FWAD is negligible.

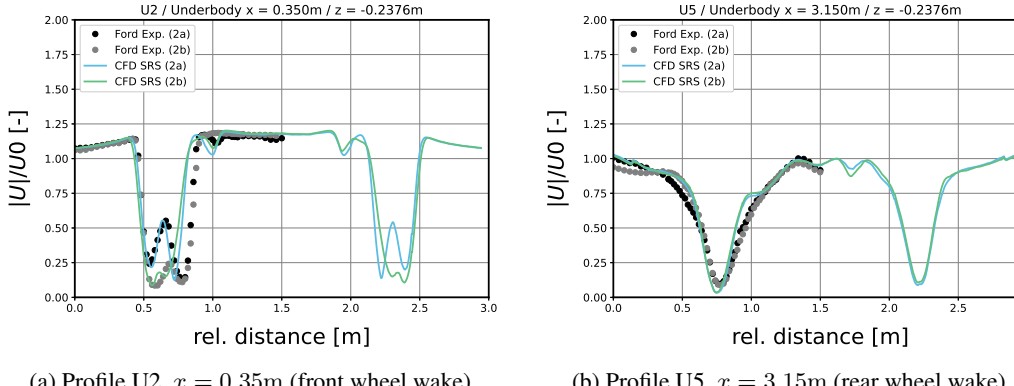

(a) Profile U2, $x = 0.35$m (front wheel wake)

(b) Profile U5, $x = 3.15$m (rear wheel wake)

Figure 21: Profiles of normalised velocity magnitude at lateral profiles between underbody and ground for cases 2a and 2b.

Fig. 21 shows comparisons of the time-averaged normalised velocity along different lateral sample lines directly behind the front and rear wheels For profile U2 (behind the front wheels), the influence of the FWAD is pronounced. At this position, the width of the wake behind the wheel is similar for both cases, but the velocity deficit is more pronounced for case 2b with the FWAD. The shape of the wake profile is also altered. The wheel-wake features two velocity minima separated by a local maximum. The FWAD reduces the local maximum, and the absolute minimum switches from the inboard to the outboard minimum. These differences in the wake behaviour are qualitatively similar in both simulations and experiment. The inboard shear layer is however further outboard, giving a narrower wake behind the front wheel in the CFD. From Fig. 21b, it can be seen that the influence of the FWAD is much less noticeable behind the rear wheels. While the agreement for the U5 profiles is again promising and the velocity deficit is in close agreement, the small differences between cases 2a and 2b in the outboard shear layer are not captured by the CFD.

In Fig. 22a, time-averaged pressure coefficient values are compared for probe position on the sidewall of the vehicles. It can be observed that the FWAD locally increases the static pressure just downstream of the front wheelhouse, and the pressure levels return to those of the configuration without FWAD within a short distance downstream. This is seen qualitatively in both experiment and CFD, however the absolute pressure values are not in perfect agreement with the experiment directly behind the front wheel ($x \approx 0.5\,\mathrm{m}$), where the static pressure is higher in the CFD. Finally, Fig. 22b shows some pressure probes located inside the left front wheelhouse, which is another challenging area for CFD due to its highly unsteady turbulent flow. For most probe positions, the FWAD causes a static pressure decrease, which is predicted in both CFD and experiment.

In summary, the employed numerical methodology delivers an encouraging agreement with experiment for the main flow features. The integral forces however show significant deviations, particularly for the absolute lift with a deviation of around 50 counts attributed to the front lift. This is believed to be, at least partly, a consequence of the coarser time step deployed to limit computational cost, a hypothesis supported by initial testing with a finer time step. The integral force deviations are furthermore found to occur also in the overwhelming majority of existing CFD data from the 3rd AutoCFD workshop.

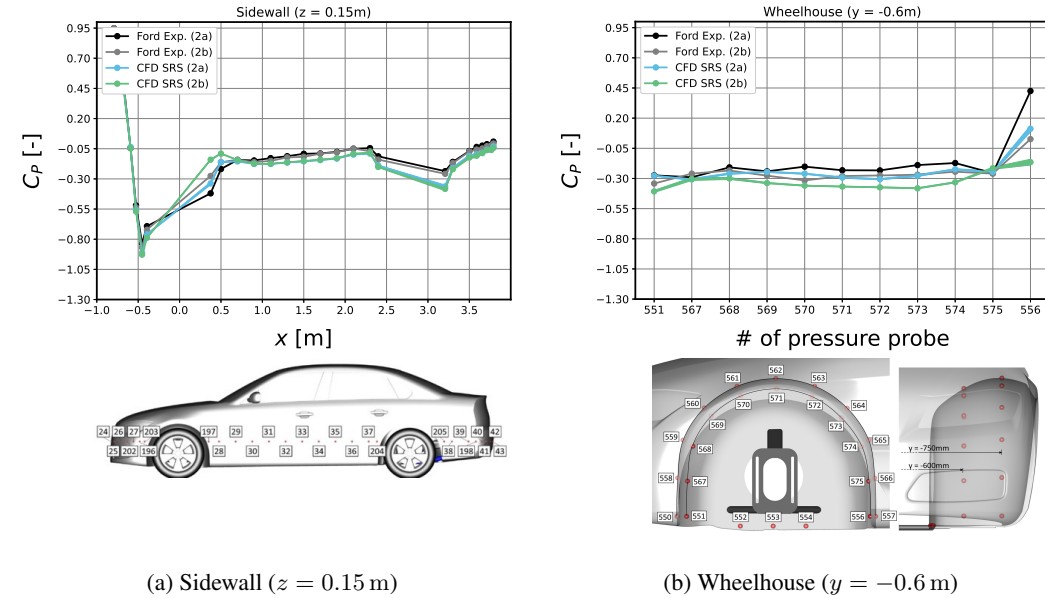

(a) Sidewall ($z = 0.15\,\mathrm{m}$)      (b) Wheelhouse ($y = -0.6\,\mathrm{m}$)

Figure 22: Comparison of time-averaged static pressure coefficient $\overline{C}_p$ for probe positions on the sidewall (a) and in the front left wheelhouse (b) between cases 2a and 2b. The lower figures of probe positions reproduced with permission from (9).

Local flow quantities generally show a very good agreement with the experimental benchmark data in the most challenging flow areas. Trends between the two configurations are generally well captured in most regions. In particular, the effect of the front-wheel air deflector on the flow around the front wheels is in very good agreement. The strongest deviations to the experimental data occur for the vehicle wake profiles. Here, the comparison with the CFD datasets from the 3rd AutoCFD workshop strongly suggests a more systematic issue in the comparability of experimental and CFD setups. In particular, the presence of the wind tunnel collector, not present in the open-road CFD domain, is expected to influence the nearby wake region. Overall, it is demonstrated that the employed scale-resolving simulation approach is capable of producing high quality dataset results.

## D    DATASET DESCRIPTION

### D.1    ACCESS TO DATASET

The dataset is openly accessible without any additional costs and is hosted on Amazon Web Services (AWS) using the Amazon Simple Storage Service (S3). The dataset is hosted within a S3 bucket located in the us-east-1 region (North Virginia region of the United States), thus additional latency can be expected for downloads to locations away from this geographical area.

The dataset README.txt will be kept up to date for any changes to the dataset and can be found at the following URL:

```
https://xxxxxxx.s3.us-east-1.amazonaws.com/drivaer/dataset/README.
txt
```

The dataset itself can be downloaded via the AWS Command Line Interface (CLI) tool, which is free-of-charge. An instruction about how to install the AWS CLI tool is given here: `https://docs.aws.amazon.com/cli/latest/userguide/getting-started-install.html`. After installing AWS CLI, you can use the following examples to download the dataset or subsets of it, given the full dataset is approximately 22TB.

Note: If you don't have an AWS account you will need to add –no-sign-request within your AWS command i.e aws s3 cp –no-sign-request –recursive etc...

**Example 1: Download all files (~22 TB)**

```
aws s3 cp --recursive s3://xxxxx/drivaer/dataset .
```

**Example 2: Only download select files (STL, images & force and moments):**

Create the following bash script that could be adapted to loop through only select runs or to change to download different files e.g boundary/volume:

```bash
#!/bin/bash

# Set the S3 bucket and prefix
S3_BUCKET="xxxxxx"
S3_PREFIX="drivaer/dataset"

# Set the local directory to download the files
LOCAL_DIR="./drivaer_data"

# Create the local directory if it doesn't exist
mkdir -p "$LOCAL_DIR"

# Loop through the run folders from 1 to 500
for i in $(seq 1 500); do
    RUN_DIR="run_$i"
    RUN_LOCAL_DIR="$LOCAL_DIR/$RUN_DIR"

    # Create the run directory if it doesn't exist
    mkdir -p "$RUN_LOCAL_DIR"

    # Download the drivaer_i.stl file
    aws s3 cp "s3://$S3_BUCKET/$S3_PREFIX/$RUN_DIR/drivaer_$i.stl" \
    "$RUN_LOCAL_DIR/" --only-show-errors

    # Download the force_mom_i.csv file
    aws s3 cp "s3://$S3_BUCKET/$S3_PREFIX/$RUN_DIR/force_mom_$i.csv" \
    "$RUN_LOCAL_DIR/" --only-show-errors
```

```
        aws s3 cp --recursive "s3://$S3_BUCKET/$S3_PREFIX/$RUN_DIR/images" \
        "$RUN_LOCAL_DIR/images/" --only-show-errors
    done
```

## D.2 LONG-TERM HOSTING/MAINTENANCE PLAN

The data is hosted on Amazon S3 for several reasons. Firstly it provides 11 nines of durability, i.e the data has very low risk of not being available globally from a technical point of view. Secondly, the data transfer speed can be as high as 300MB/s due to the high bandwidth Amazon network (the bottleneck will likely be on the user rather than Amazon), since a poor server performance could mean extremely long download times given the size of the dataset. Thirdly, no account or credentials are required to download the data (only AWS CLI tools, described above, which are free to download and use). Finally, there are very limited number of providers that can host such large datasets ( 22TB) and make them available in such a manner. Over the coming year we will identify additional providers to mirror this dataset, to further increase the availability in the very unlikely scenario there was a problem with the Amazon S3 storage option. In addition, a dedicated website will be created for the AhmedML (2), WindsorML (1) and DrivAerML datasets to help further clarify where the data is hosted and to communicate any additional mirroring sites.

## D.3 LICENSING TERMS

The dataset is provided with the Creative Commons CC-BY-SA v4.0 license[12]. The license grants the user the right to **share** the work, e.g. by copying and redistributing the material in any medium or format for any purpose, which includes redistribution for commercial purposes. Likewise, the material can be **adapted** by remixing or transforming it, or building upon the material for any purpose. In case of redistribution, you must give appropriate credit to the original authors, which includes providing the names of the creators and attribution parties, a copyright notice, a license notice, a disclaimer notice, and a link to the material. You must also indicate if you modified the material and retain an indication of previous modifications. You may do so in any reasonable manner, but not in any way that suggests the licensor endorses you or your use ("Attribution" clause). If you remix, transform, or build upon the material, you must distribute your contributions under the same license as the original ("ShareAlike" clause). No warranties are given. The license may not give you all of the permissions necessary for your intended use. For example, other rights such as publicity, privacy, or moral rights may limit how you use the material. A full description of the license terms is provided under the following URL:

```
https://xxxxxxx.s3.us-east-1.amazonaws.com/drivaer/dataset/
LICENSE.txt
```

## D.4 INTENDED USE & POTENTIAL IMPACT

The dataset was created with the following intended uses:

- Development and testing of data-driven ML surrogate models (e.g meshGraphNet (5)) for the prediction of external aerodynamics quantities (lift, drag, pressure, velocity) on road car geometries of the notchback type.

- Testing of physics-driven ML approaches on a complex set of geometries and test conditions.

- For academia, a stepping-stone dataset after more fundamental, 'simpler' datasets (e.g AhmedML (2) and WindsorML (1)). For an automotive company, it can be a useful dataset that is similar in size and complexity to an internal non-public dataset, i.e an automotive company's own data.

- As a 'challenge' test-case at future conferences/workshops to benchmark the performance of different ML approaches for an open-source automotive dataset.

---

[12]https://creativecommons.org/licenses/by-sa/4.0/deed.en

- The dataset was created for the AutoCFD4 [13] workshop, as a test case for the AI/ML technical working group, to allow for the assessment of different ML approaches on an identical open-source dataset.
- Large-scale dataset for the study of flow physics over road-cars, i.e potential non-ML use-case.

The potential impact could be:

- Establishing an industry-standard benchmark for the testing of ML methods for the automotive external aerodynamics community.
- Allowing for fairer testing of large-scale CFD versus ML approaches, i.e training and inference time on non-canonical problems.
- Addressing the lack of high-quality, public-domain training data, thereby fostering innovation in ML for automotive aerodynamics.

### D.5 DOI

At present there is no specific DOI for the dataset itself, given Amazon S3 is not a resource that can easily be assigned a DOI. However it is the intention of the authors to create a DOI once a suitable mechanism is found. For the time-being, users of the dataset will be encouraged to cite this dataset paper.

### D.6 DETAILS OF PROVIDED DATA

In the dataset, each folder (e.g `run_1`, `run_2`, ..., `run_i`, etc.) corresponds to a different geometry, where "i" is the run number that ranges from 1 to 500. All run folders feature the same structure:

```
run_i/
|
|- boundary_i.vtp
|- drivaer_i.stl
|- force_mom_i.csv
|- force_mom_constref_i.csv
|- geo_parameters_i.csv
|- geo_ref_i.csv
|- volume_i.vtu
|- images/
    |
    |- fig_runi_SRS_<Q-value>_<view>_<variable>.png
    |- fig_runi_SRS_<variable>_<slice>_<position>.png
    |- fig_runi_SRS_<surface>_<variable>.png
    |- fig_runi_SRS_<slice>_<position>_grid.png
    |- fig_runi_evolution_Cd.png
    |- fig_runi_evolution_Cl.png
    |- fig_runi_evolution_Cs.png
    |- fig_runi_solverStats_initialResidual.png
|- slices/
    |
    |- <sliceNormal>\_<position>.vtp
```

A brief description of the contents in each file, including the file format and the file size, is given in the Tab. 2. Tab. 4 provides a list of output flow variables, which were all obtaining through time-averaging of the initial-transient free portion of the unsteady flow field. In general, the dataset contains outputs of different complexity. This offers ML researchers the flexibility to train their models either via the full three-dimensional flow solution of the CFD domain or to use subsets of the solution instead:

---

[13]https://autocfd.org

- **Volume field:** The complete, three-dimensional and time-averaged flow field is provided. The most commonly analysed quantities in automotive aerodynamics were stored, including first and second order flow statistics (see Tab. 4).

- **Surface field:** The complete, time-averaged flow field on the car surface is provided. All flow quantities necessary to compute the integral force coefficients (see Sect. B) along with time-averaged surface pressure fluctuations are included.

- **Slices of the volume field:** A total of 65 two-dimensional slices through the volume mesh are provided, where the slice positions are shown in Fig. 23. The $x$-normal slices range from $x = -1.5\,\text{m}$ to $x = 6.5\,\text{m}$, the $y$-normal slices from $y = -1.4\,\text{m}$ to $y = 1.4\,\text{m}$ and $z$-normal slices from $z = -0.2\,\text{m}$ to $z = 1.4\,\text{m}$, with a step size of $0.2\,\text{m}$ in between. Three additional slice positions located at $x = 0.407\,\text{m}$, $y = 4.007\,\text{m}$ and $z = -0.2376\,\text{m}$ are included, which are specific to the post-processing conducted in the AutoCFD-4 workshop. The user of the dataset should be aware that the positions of the slices are fixed to the CFD coordinate system and are thus not adapted to the morphed geometry (e.g. by scaling the x-positions with the car length). This implies that one particular slice position does not also represent the same relative position in the flow field, e.g. an $x$-normal slice that is located in the car wake for one geometry might cut through the rear window region for another geometry.

- **Force coefficients:** Time-averaged force and moment coefficients are also provided, together with their 95% statistical confidence intervals evaluated by Meancalc. These are given with two different normalisations (see Tab. 3): One using the reference values for the car wheelbase $L_{\text{ref}}$, the frontal area $A_{\text{ref}}$ and the centre of rotation $\vec{x}_{\text{ref}}$ specific to each geometry variant, and a second using the nominal reference values of the baseline DrivAer geometry (denoted with the suffix "Ref"). Additionally, the Meancalc evaluation plots of the force coefficients described in Sect. A.3 and shown in Figs. 8 and 9 are included.

- **Flow visualisations:** Image files with contour plots of selected time-averaged flow quantities on the 2D slices and the car surface are provided for every case. They are intended to give an impression of the flow field quickly and conveniently, without having to process the raw data first. Examples of the plots for selected 2D slices and surface contours are given in Fig. 24. Care was taken to plot meaningful variable ranges for each quantity.

All provided data is either written in ASCII or in the open source format VTK (i.e. *.vtp and *.vtu). The VTK output files can be loaded in the most common 3D data visualisation tools, e.g. using the open source software ParaView[14]. The data can also be further post-processed with Python and Java scripts with the corresponding VTK extension/module.

---

[14]https://www.paraview.org/

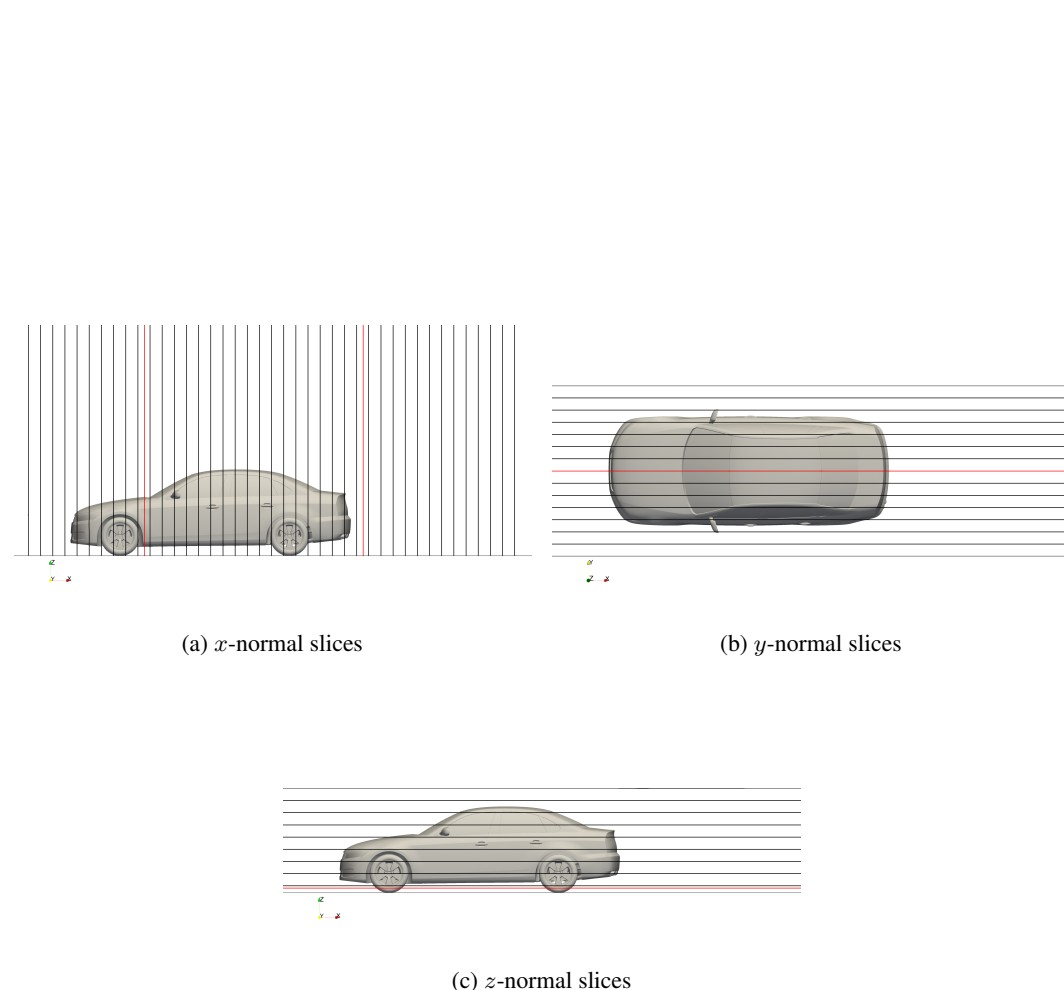

(a) $x$-normal slices

(b) $y$-normal slices

(c) $z$-normal slices

Figure 23: Positions of extracted slices in .vtp format. Slices in red correspond to the additional AutoCFD-4 workshop post-processing planes.

Table 2: Description of the main components of the dataset

| Output | Size | Format | Description |
|---|---|---|---|
| drivaer_i.stl | 135 MB | stl | surface mesh (tris) of the DrivAer geometry ($\approx$ 750k cells) |
| images/... fig_<case>_<quantity>_<view>_<sliceNormal>_<position>.png fig_<case>_<surface>-<view>_<quantity>.png | 87 MB | png | folder containing images of the flow-field |
| slices/... <sliceNormal>_<position>.vtp | 907 MB | vtp | folder containing slices of the domain volume in X, Y, Z with time-averaged flow quantities |
| boundary_i.vtp | 612 MB | vtp | time-averaged flow quantities on the DrivAer ($\approx$ 8.8M cells) |
| volume_i.vtu | 44 GB | vtu | time-averaged flow quantities within the domain volume ($\approx$ 160M cells) |
| geo_ref_i.csv | 66 KB | csv | reference values such as $A_{ref}$ and $L_{ref}$ of each geometry |
| geo_parameters_i.csv | 66 KB | csv | reference geometry values used to define the particular geometry via the DoE method |
| force_mom_i.csv | 66 KB | csv | time-averaged drag, lift, front-lift, rear-lift and side force coefficients |
| force_mom_constref_i.csv | 66 KB | csv | time-averaged drag, lift, front-lift, rear-lift and side force coefficients using constant frontal-area and moment length |

Table 3: Reference quantities used for normalisation of force and moment coefficients.

| Variable | force_mom_i.csv | force_mom_constref_i.csv |
|---|---|---|
| $U_\infty$ | 38.889 m/s | 38.889 m/s |
| $\rho_\infty$ | 1 kg/m$^3$ | 1 kg/m$^3$ |
| $A_{ref}$ | $1.779 - 2.636$ m$^2$ | 2.17 m$^2$ |
| $L_{ref}$ | $2.636 - 3.035$ m | 2.78618 m |
| $\vec{x}_{ref}$ | $\begin{bmatrix} 1.325\text{ m} \\ 0\text{ m} \\ -0.3176\text{ m} \end{bmatrix} - \begin{bmatrix} 1.5245\text{ m} \\ 0\text{ m} \\ -0.3176\text{ m} \end{bmatrix}$ | $\begin{bmatrix} 1.40009\text{ m} \\ 0\text{ m} \\ -0.3176\text{ m} \end{bmatrix}$ |

Table 4: List of output quantities in the provided dataset files, all quantities are time-averaged.

| Symbol | Units | Field name | Description |
|--------|-------|------------|-------------|
| **volume_i.vtu** | | | |
| $\overline{p^*}$ | $[\mathrm{m}^2/\mathrm{s}^2]$ | `pMeanTrim` | relative kinematic pressure |
| $\overline{(p^{*\prime})^2}$ | $[\mathrm{m}^4/\mathrm{s}^4]$ | `pPrime2MeanTrim` | square of mean pressure fluctuations |
| $\overline{U_i}$ | $[\mathrm{m/s}]$ | `UMeanTrim` | velocity vector |
| $\overline{u_i' u_j'}$ | $[\mathrm{m}^2/\mathrm{s}^2]$ | `UPrime2MeanTrim` | resolved Reynolds stress tensor |
| $\overline{R_{ij}}$ | $[\mathrm{m}^2/\mathrm{s}^2]$ | `turbulenceProperties:RMeanTrim` | modelled Reynolds stress tensor |
| $\overline{\nu_t}$ | $[\mathrm{m}^2/\mathrm{s}]$ | `nutMeanTrim` | turbulent eddy viscosity |
| $\overline{C_p}$ | $[-]$ | `CpMeanTrim` | static pressure coefficient |
| $\overline{C_{pt}}$ | $[-]$ | `CptMeanTrim` | total pressure coefficient |
| $\lvert\overline{U_i}\rvert/U_\infty$ | $[-]$ | `magUMeanTrimNorm` | normalised velocity magnitude |
| $\overline{C_{\mathrm{dl}}}$ | $[-]$ | `microDragMeanTrim` | micro drag coefficient |
| **slices/<sliceNormal>_<position>.vtp** | | | |
| $\overline{p^*}$ | $[\mathrm{m}^2/\mathrm{s}^2]$ | `pMeanTrim` | relative kinematic pressure |
| $\overline{(p^{*\prime})^2}$ | $[\mathrm{m}^4/\mathrm{s}^4]$ | `pPrime2MeanTrim` | square of mean pressure fluctuations |
| $\overline{U_i}$ | $[\mathrm{m/s}]$ | `UMeanTrim` | velocity vector |
| $\overline{u_i' u_j'}$ | $[\mathrm{m}^2/\mathrm{s}^2]$ | `UPrime2MeanTrim` | resolved Reynolds stress tensor |
| $\overline{\nu_t}$ | $[\mathrm{m}^2/\mathrm{s}]$ | `nutMeanTrim` | turbulent eddy viscosity |
| $\overline{C_p}$ | $[-]$ | `CpMeanTrim` | static pressure coefficient |
| $\overline{C_{pt}}$ | $[-]$ | `CptMeanTrim` | total pressure coefficient |
| $\lvert\overline{U_i}\rvert/U_\infty$ | $[-]$ | `magUMeanTrimNorm` | normalised velocity magnitude |
| $\overline{C_{\mathrm{dl}}}$ | $[-]$ | `microDragMeanTrim` | micro drag coefficient |
| **boundary_i.vtp** | | | |
| Symbol | Unit | Field name | Description |
| $\overline{p^*}$ | $[\mathrm{m}^2/\mathrm{s}^2]$ | `pMeanTrim` | relative kinematic pressure |
| $\overline{(p^{*\prime})^2}$ | $[\mathrm{m}^4/\mathrm{s}^4]$ | `pPrime2MeanTrim` | square of mean pressure fluctuations |
| $\overline{\tau_i}$ | $[\mathrm{m}^2/\mathrm{s}^2]$ | `wallShearStressMeanTrim` | wall shear stress vector |
| $\overline{C_p}$ | $[-]$ | `CpMeanTrim` | static pressure coefficient |

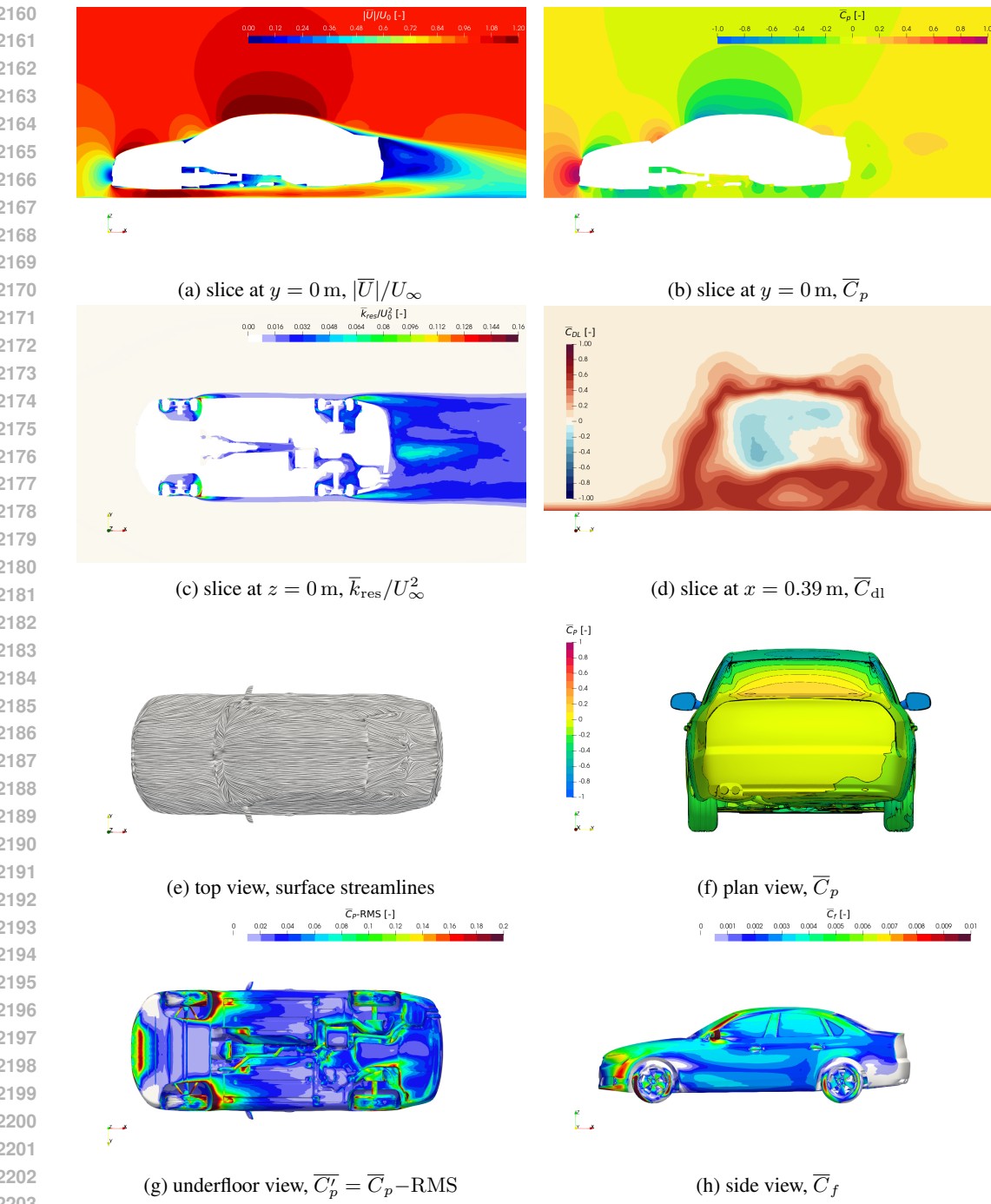

(a) slice at $y = 0\,\mathrm{m}$, $|\overline{U}|/U_\infty$

(b) slice at $y = 0\,\mathrm{m}$, $\overline{C}_p$

(c) slice at $z = 0\,\mathrm{m}$, $\overline{k}_{\mathrm{res}}/U_\infty^2$

(d) slice at $x = 0.39\,\mathrm{m}$, $\overline{C}_{\mathrm{dl}}$

(e) top view, surface streamlines

(f) plan view, $\overline{C}_p$

(g) underfloor view, $\overline{C'_p} = \overline{C}_p - \mathrm{RMS}$

(h) side view, $\overline{C}_f$

Figure 24: Examples of provided contour and surface plots. Selected images of validation case 2a are shown.

### D.7 Geometry variants

500 geometric variations of the DrivAer notchback were created to replicate as closely as possible the potential range of design studies that an automotive company would create for this category of vehicle. This differs from the AhmedML (2) and WindsorML (1) datasets, where more freedom was given to adapt the geometry more strongly given the academic nature of the geometry and to test the ability of a model to predict a wide range of geometries. This DrivAerML dataset is designed to test the ability of a ML model to predict subtle and small changes in the geometry, which is more representative of industry use of CFD for product design.

To achieve this a set of morphing boxes was constructed around the baseline DrivAer Notchback using the ANSA software of BETA-CAE Systems[15], (see Fig. 25a), allowing geometry variants to be created in a systematic manner. The morphing box topology prevents undesired distortions (e.g. the wheels remain circular when the vehicle length is stretched). The parameters for morphing the baseline geometry are listed together with their ranges in Tab. 5, which were chosen to avoid unrealistic shapes based on engineering judgement. The range of geometries is intended to produce different flow topologies and to test the generalisability of ML approaches, but within a typical engineering process.

Table 5: Geometry parameters and limits for the DrivAerML model. Note that parameters are defined as changes relative to the baseline geometry.

| Parameter | Min | Max |
|---|---|---|
| Vehicle_Length | -150 mm | +200 mm |
| Vehicle_Width | -100 mm | +100 mm |
| Vehicle_Height | -100 mm | +100 mm |
| Front_Overhang | -150 mm | +100 mm |
| Front_Planview | -75 mm | +75 mm |
| Hood_Angle | -50 mm | +50 mm |
| Approach_Angle | -40 mm | +30 mm |
| Windscreen_Angle | -150 mm | +150 mm |
| Greenhouse_Tapering | -100 mm | +100 mm |
| Backlight_Angle | -100 mm | +200 mm |
| Decklid_Height | -50 mm | +50 mm |
| Rearend_tapering | -90 mm | +70 mm |
| Rear_Overhang | -150 mm | +100 mm |
| Rear_Diffuser_Angle | -50 mm | +50 mm |
| Vehicle_Ride_Height | -50 mm | +50 mm |
| Vehicle_Pitch (positive nose up) | -1° | +1° |

In order to ensure optimal coverage of the design space, a design of experiments (DoE) tool in ANSA was used to create the parametric values for 500 experiments using a Modified Extensible Lattice Sequence algorithm, which fills the parameter space evenly, also for subsets of and extensions to the dataset. Figure 26 shows an example of how the distribution of points is optimally spread through the parameter space for two of the 16 parameters (rear end tapering and backlight angle).

Figures 27 & 28 show the variation of drag, lift, front-lift & rear-lift coefficients for each run, the former for constant $A_{ref}$ and $L_{ref}$ and the latter with these quantities varied for each geometry. Providing both outputs was intentional, since a constant reference area and length are sometimes used in the industry. Once multiplied by the area, it gives a sense of the actual force, e.g. a car with a larger frontal area will most likely produce in absolute terms a larger drag force. However, if one wants to focus on the aerodynamic efficiency of the vehicle (which is ultimately what the drag or lift coefficient expresses in practice), the effect of shape changes should be separated from simply changing the size of a given shape. In this scenario it is better to use a reference area and length dependant on the specific geometry. The outcome is that the spread (i.e. between minimum and maximum) of force and moment coefficients is larger when a constant reference area and length are

---

[15]https://www.beta-cae.com/ansa.htm

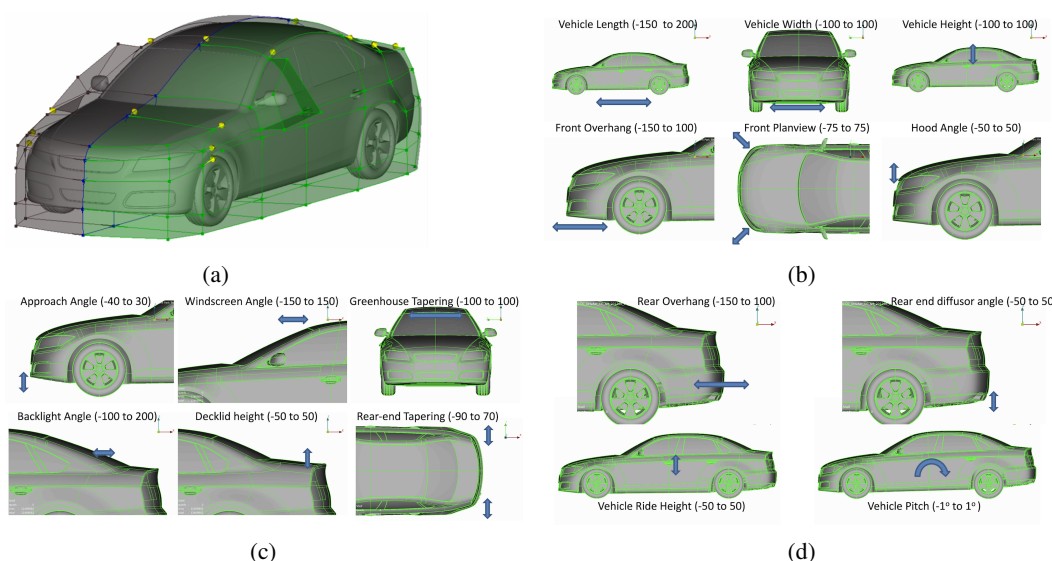

Figure 25: Generation of geometry variants based on the baseline DrivAer model. Visualisation of ANSA morphing boxes (a), visualisation of the 16 design parameters (b-d).

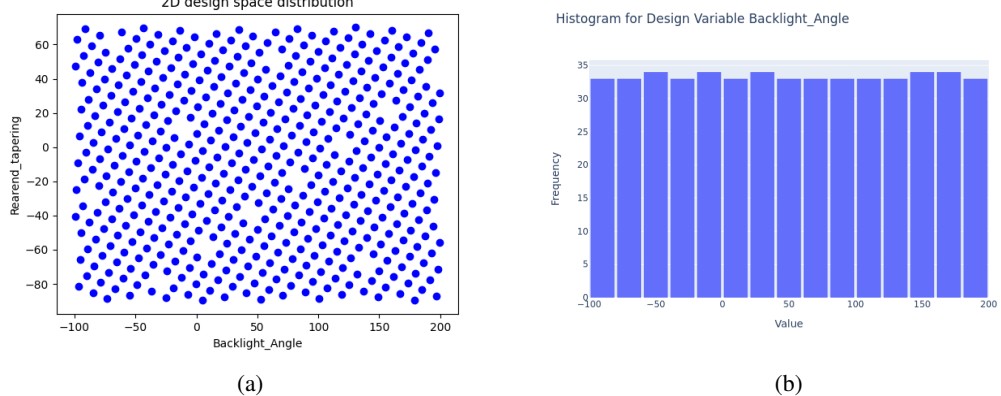

Figure 26: Visualisation of sample distribution in 2D design space, exemplary for the parameters *Backlight Angle* and *Rearend Taper* (a). Histogram of parameter values for design parameter *Backlight Angle* (b).

used (Fig. 27) compared to when the geometry-specific reference quantities are used (Fig. 28). From a ML perspective it is unclear which variant is preferable, thus both were provided.

Regardless of the choice of normalisation, the spread of drag and lift values covers a range that would be considered large in the automotive community, i.e. a drag coefficient of 0.24 would be considered a highly aerodynamically optimised vehicle, whereas $C_d = 0.32$ would be considered inefficient. Thus from a practical point of view, the range of force coefficients indicates that the geometry variants have created a diversity of designs representative of industrial automotive aerodynamics.

To give an initial impression of the effect of some geometry parameters on the aerodynamic force coefficients, Fig. 29 shows some examples where particularly clear trends emerge. The drag coefficient tends to increase with positive (nose-up) vehicle pitch, as seen in the top row of the figure. Increasing the vehicle width correlates with increasing lift coefficient, perhaps due to growth in the surface area of upward-facing surfaces where negative pressure coefficient dominates (e.g. bonnet and roof, c.f. Fig. 17a). Decreasing the approach angle (i.e. moving the underfloor leading edge downwards) is seen to increase front downforce (decrease front lift), since doing so is expected to intensify

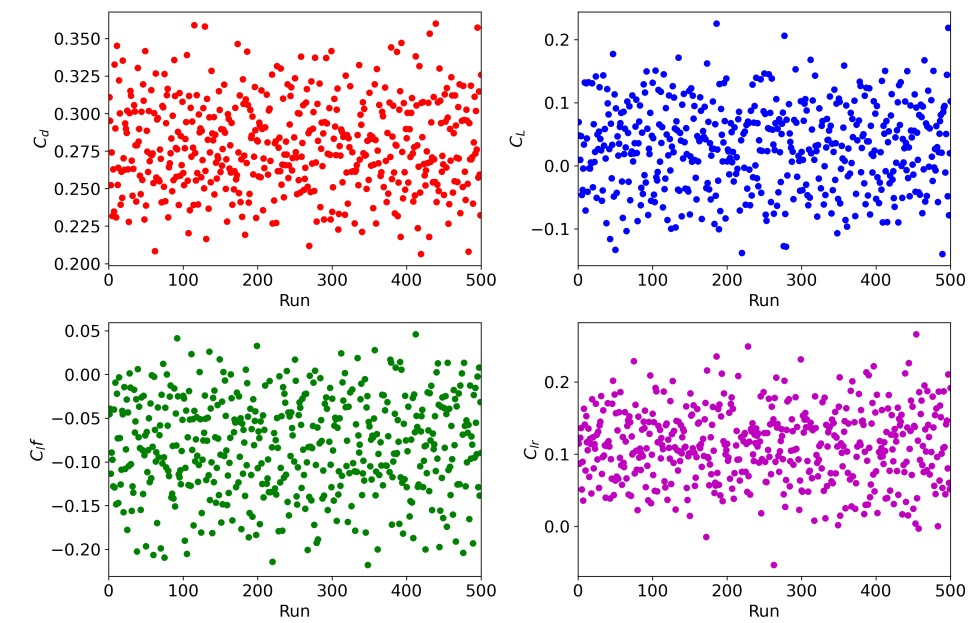

Figure 27: Variation of different force coefficients against geometry design using constant $A_{ref}$ and $L_{ref}$

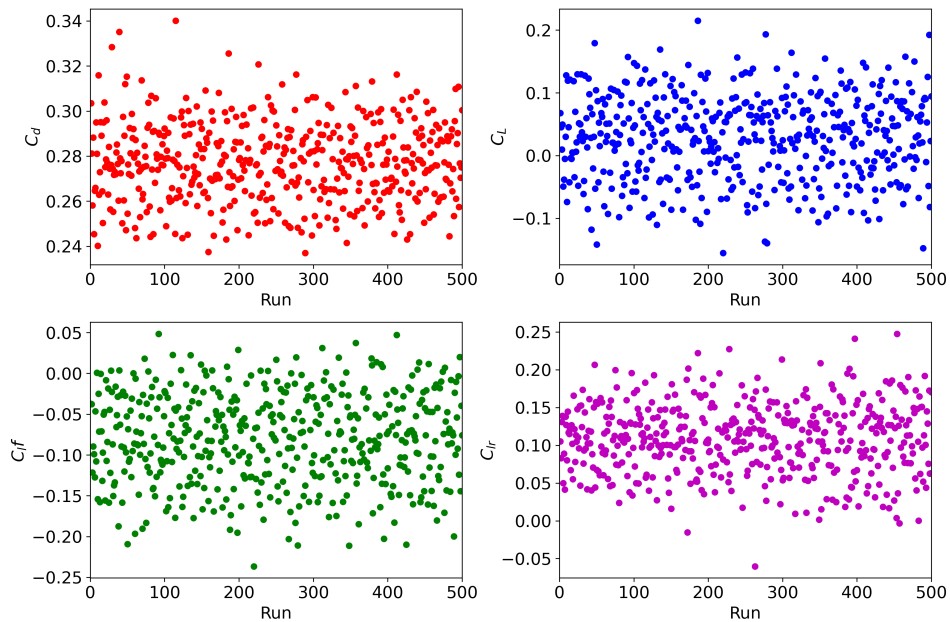

Figure 28: Variation of different force coefficients against geometry design using $A_{ref}$ and $L_{ref}$ calculated per geometry

the suction peak here. Finally, upward movement of the deck lid is associated with increased rear downforce (reduced rear lift). This also makes sense aerodynamically: Considering the car as a lifting body, the change reduces its effective camber, thereby reducing lift.

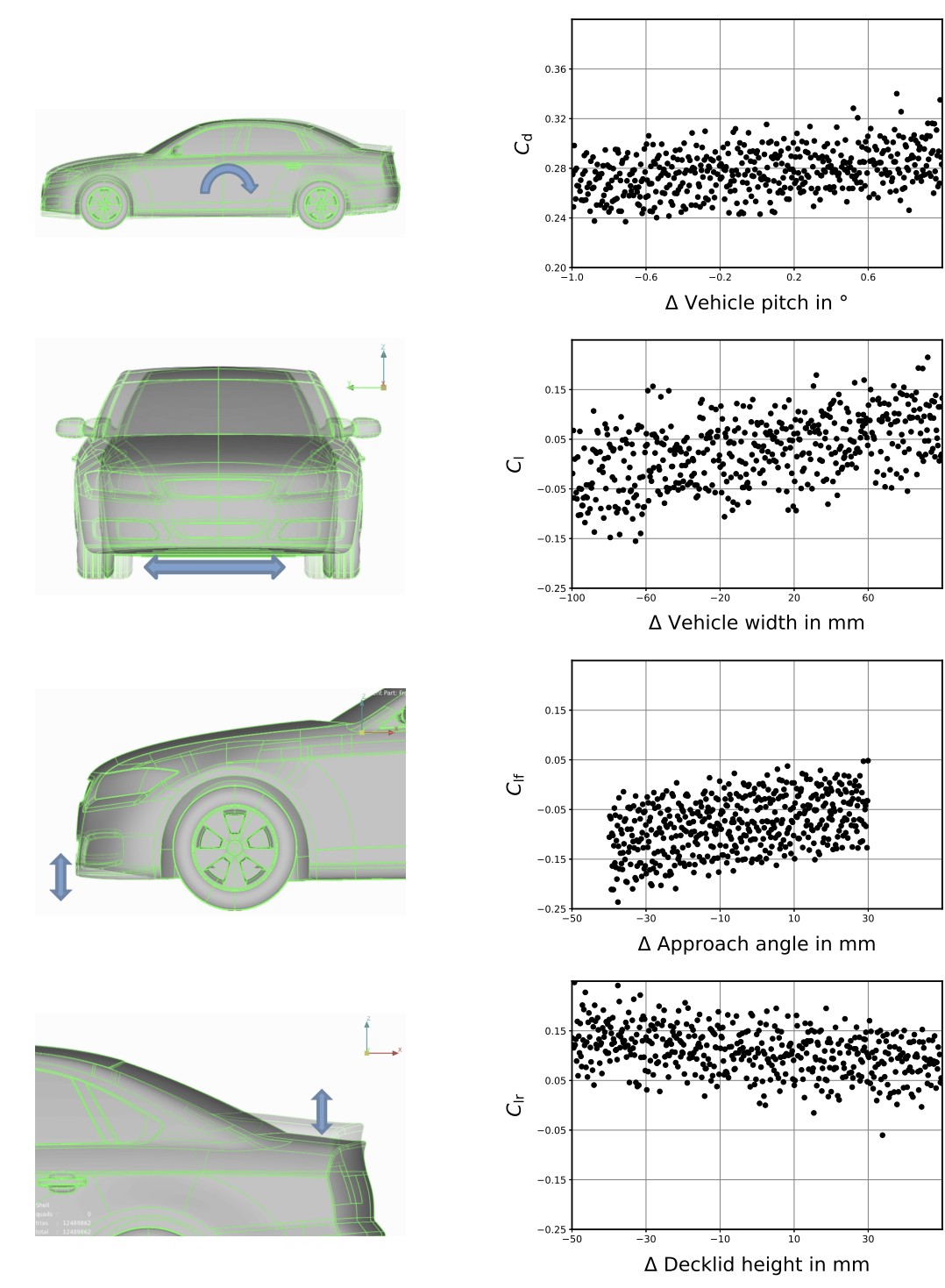

Figure 29: Variation of force coefficients with selected geometry parameters over all 500 samples in the dataset. The force coefficients (plotted in the right-hand column) are normalised with the individual reference area of each variant. The left-hand column visualises the geometry parameter by superimposing transparent images of the min/max range values.

### D.7.1 FLOW FIELDS

To illustrate the difference between a high drag geometry and a low drag geometry, we show a range of post-processing outputs for two such runs (run 115 with $C_d = 0.340$ and run 289 with $C_d = 0.237$) in Fig. 30. The main difference in the geometry is the length and width, where the shorter, wider vehicle results in a larger wake and a larger drag coefficient (even when non-dimensionalising for the larger frontal area).

However, whereas these two examples show a clear difference in flow field, the differences in the flow fields are in general much more subtle, as illustrated in Figures 31, 32, 33, 34 & 35 for runs 1 to 18.

For example the mean skin-friction in Figures 32 & 33 indicate largely similar separation patterns despite large changes in drag coefficient (c.f. Fig. 28), which suggests a cumulative effect of smaller drags rather than dramatic changes in flow separation from a single location. This is potentially a useful test for the capability of ML methods to predict these smaller changes compared to a large abrupt change in geometry and flow field.

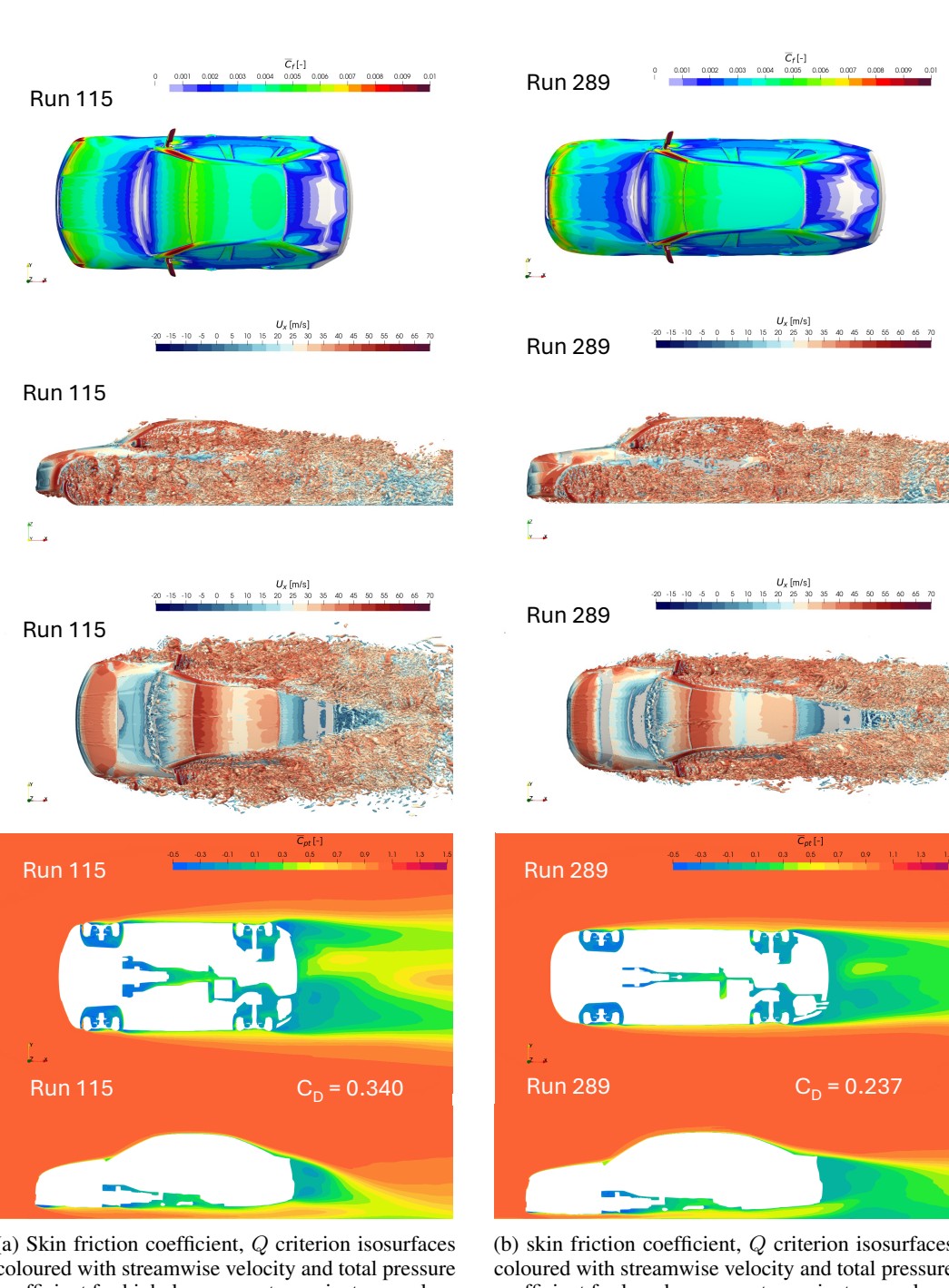

(a) Skin friction coefficient, $Q$ criterion isosurfaces coloured with streamwise velocity and total pressure coefficient for high drag geometry variant example

(b) skin friction coefficient, $Q$ criterion isosurfaces coloured with streamwise velocity and total pressure coefficient for low drag geometry variant example

Figure 30: Variation of flow fields for high-drag (left) and low-drag (right) designs

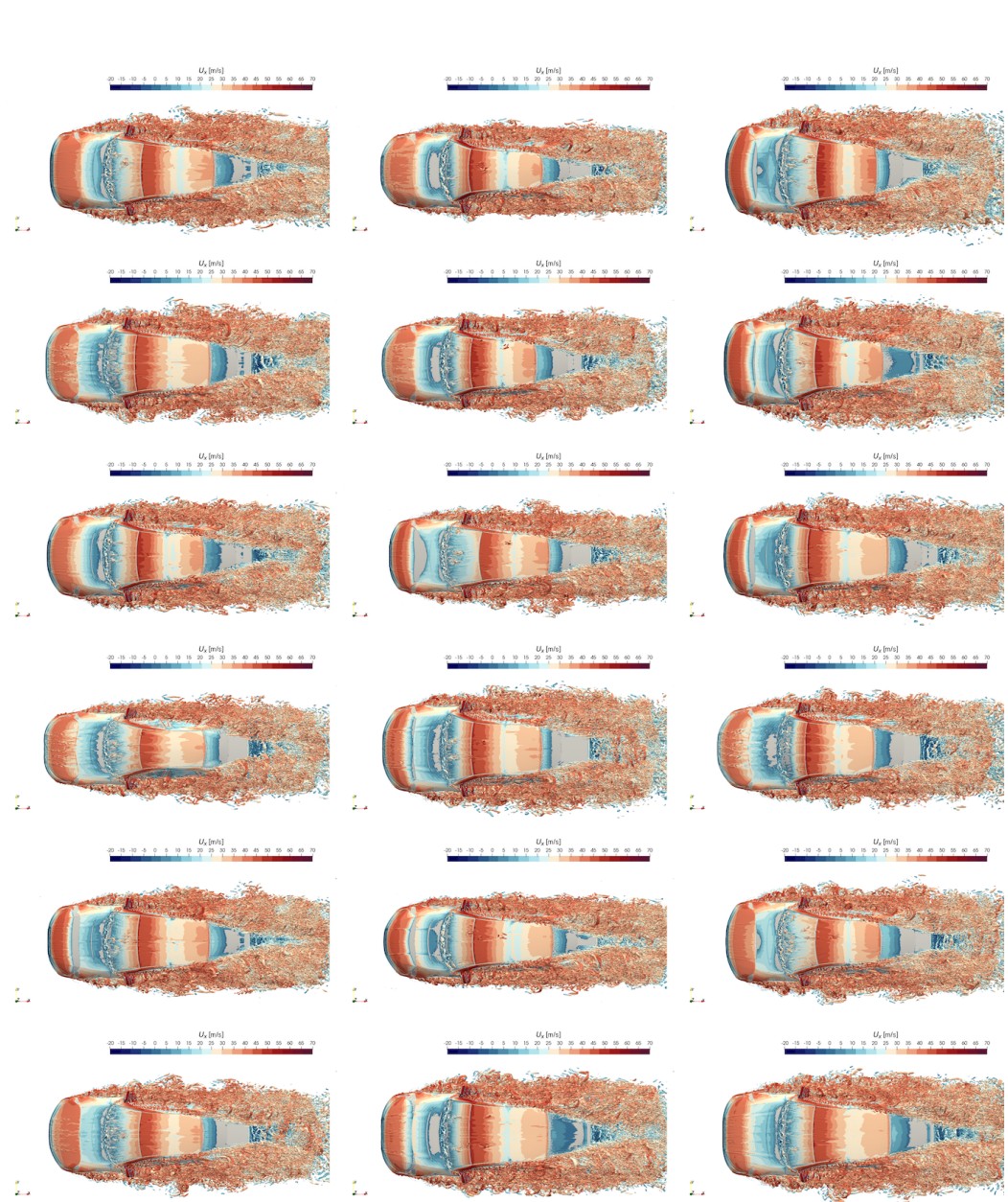

Figure 31: Isosurfaces of the $Q$ criterion (coloured by streamwise velocity) for runs 1 to 18

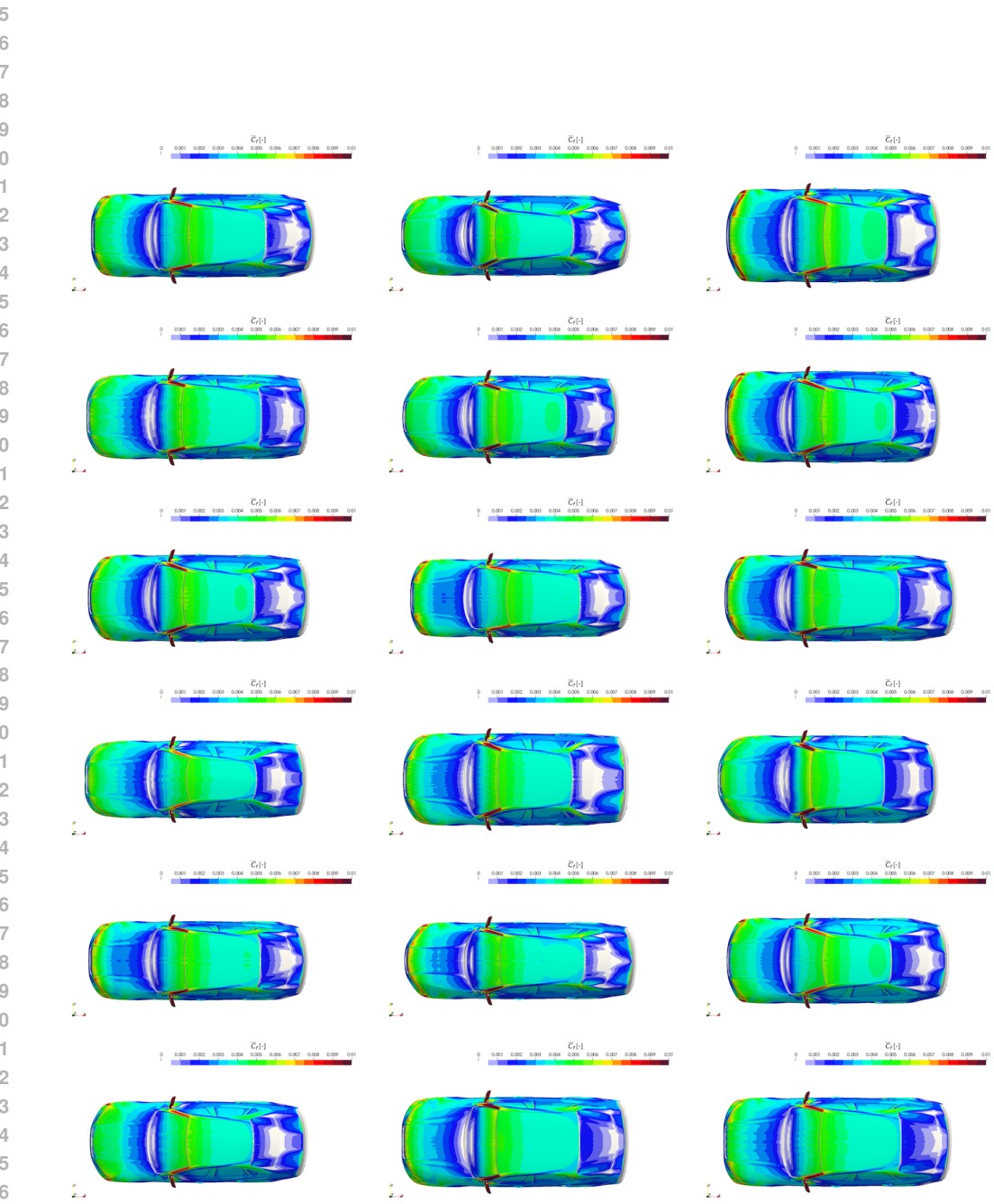

Figure 32: Skin friction contours for runs 1 to 18

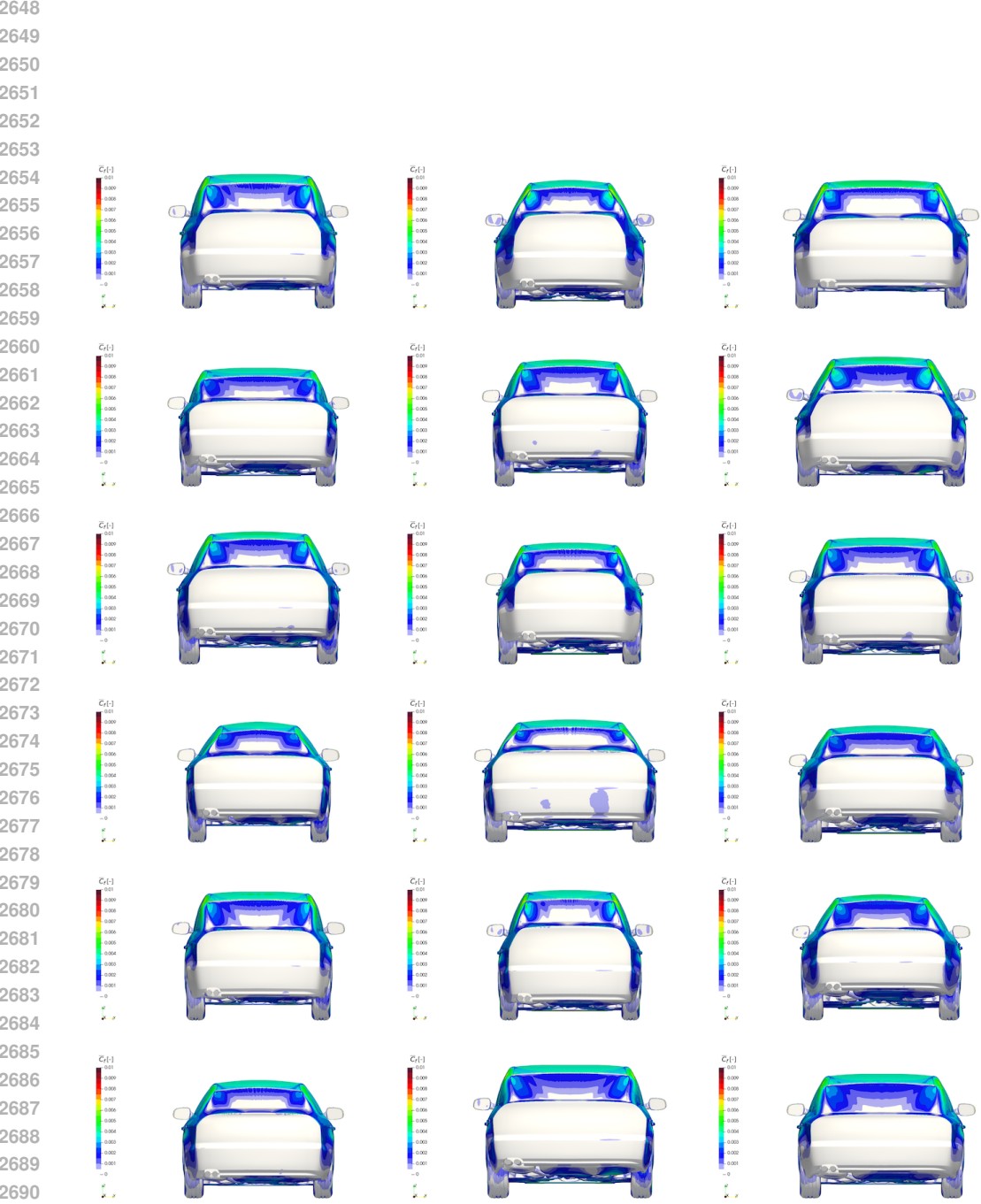

Figure 33: Skin friction contours for runs 1 to 18 from rear camera angle

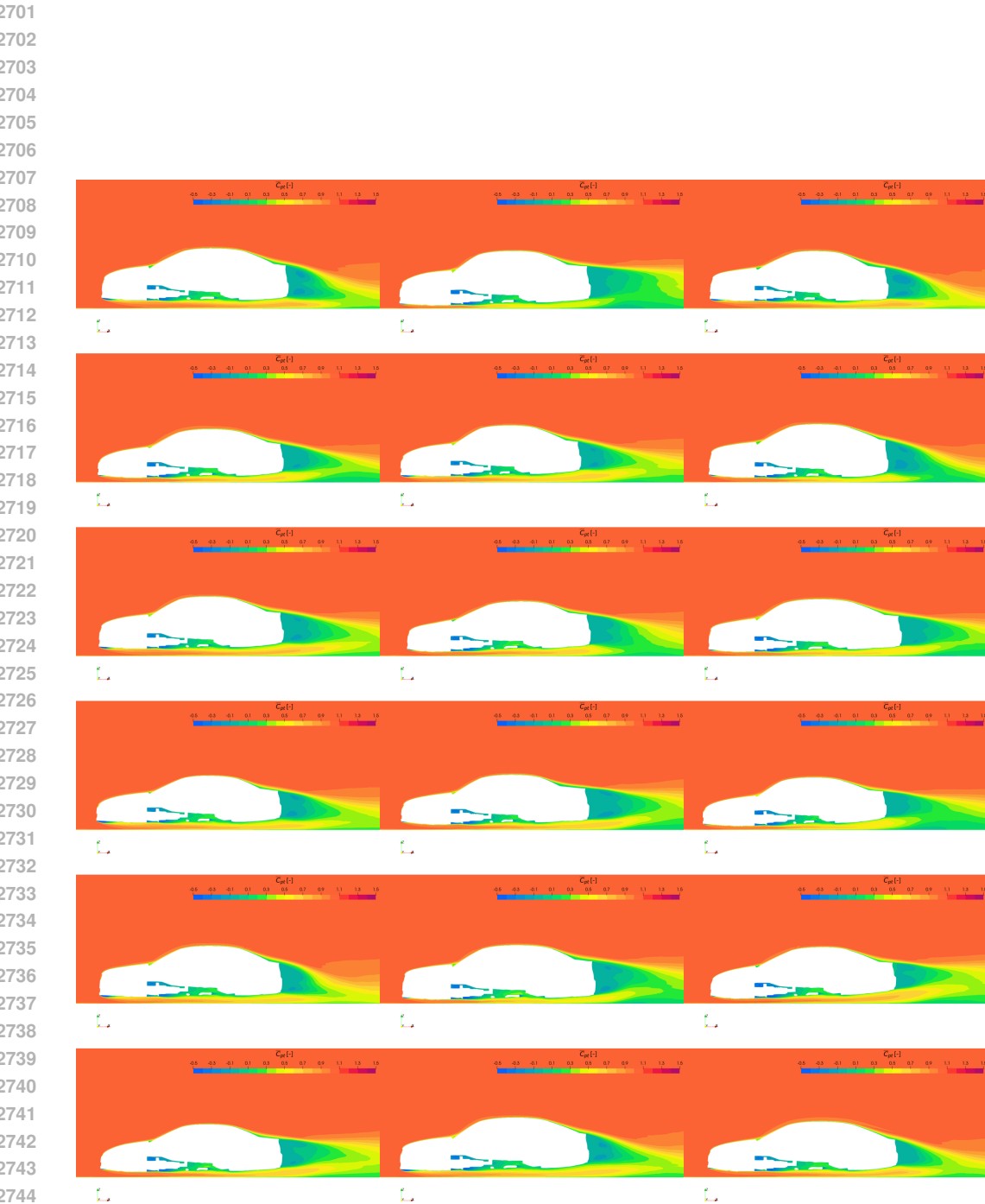

Figure 34: Total pressure coefficient at $y = 0$ plane for runs 1 to 18

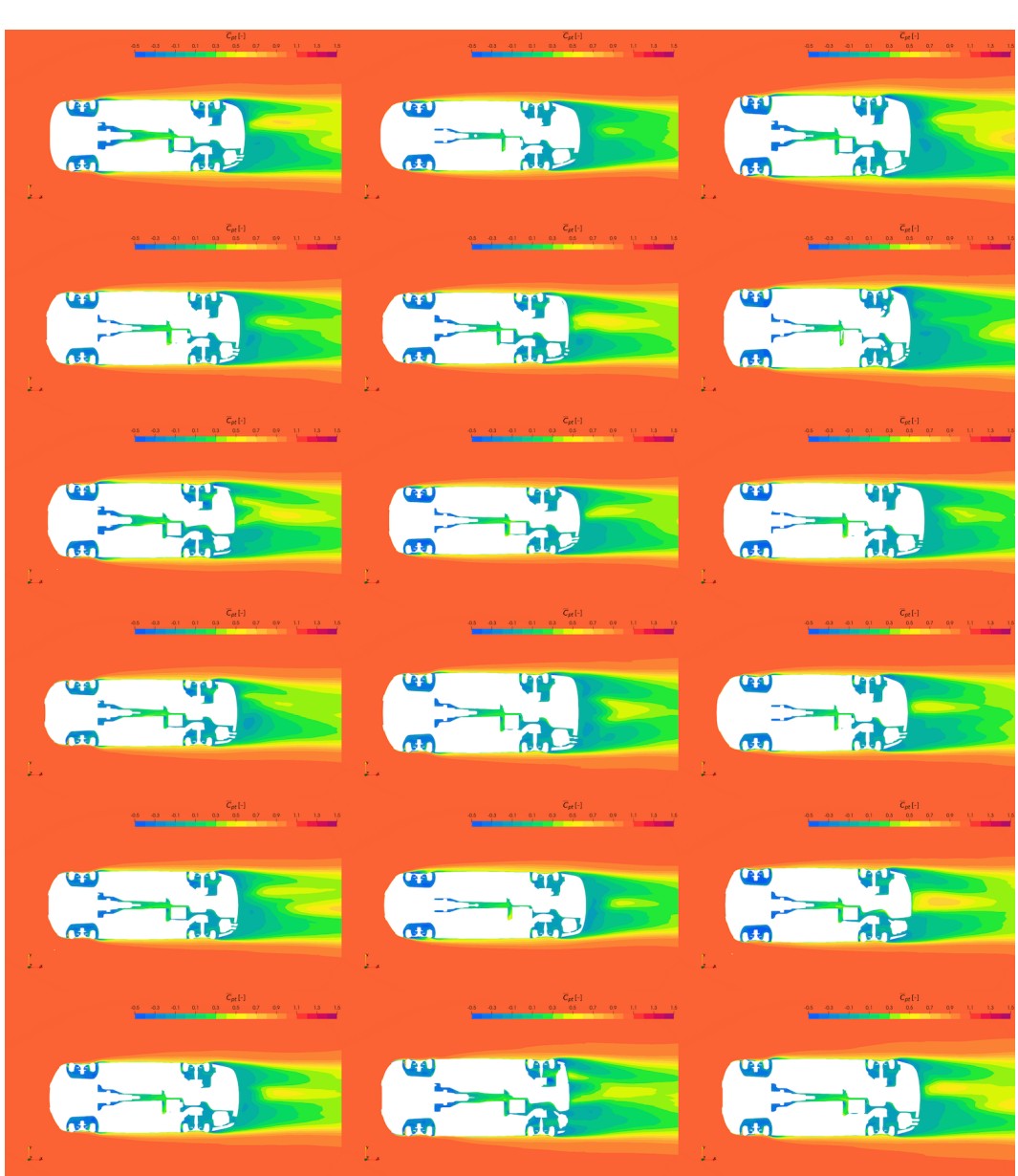

Figure 35: Total pressure coefficient at $z = 0$ plane for runs 1 to 18

# E  ML EVALUATION

## E.1  SUMMARY

We have conducted preliminary analysis on our dataset using a modified version of one of the state-of-the-art scientific machine learning (SciML) methods, MeshGraphNet (5) on various tasks to illustrate the practicality of the dataset for ML evaluation. We utilize the encoder-processor-decoder architecture in MeshGraphNets and modify the method to enable it to make time-averaged predictions (2).

The entire drivaer dataset is split into training (60%), validation (20%), and test (20%) sets. For each use case, the model is trained on the training set, and the checkpoint that had the best validation error was used to obtain the inference results on the test set.

For the DrivAer dataset, using the predicted surface pressure and wall-shear stress on the 8M node vtp surface mesh, we obtain predictions (shown in Figure 36) for the drag coefficient with a mean absolute percentage error (MAPE) of 0.032 and a mean absolute error (MAE): 0.009. For the lift coefficient the mean absolute error (MAE): 0.0164. The surface contours of the actual, predicted and error for the mean pressure and wall-shear stress are shown in Figure 37. Training time is approximately 108 hours on x8 NVidia L40s GPUs and the inference time is less than a minute on the same hardware. Please note that these runs are preliminary and further work to optimize the methodology and hyperparameters is on-going which will published in future papers.

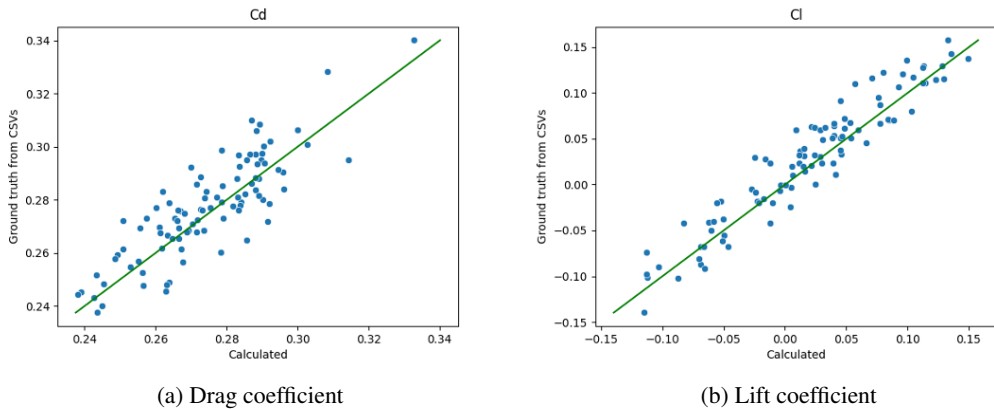

(a) Drag coefficient                    (b) Lift coefficient

Figure 36: Actual vs predicted for the force coefficients obtained through integration of the wall-shear stress and pressure

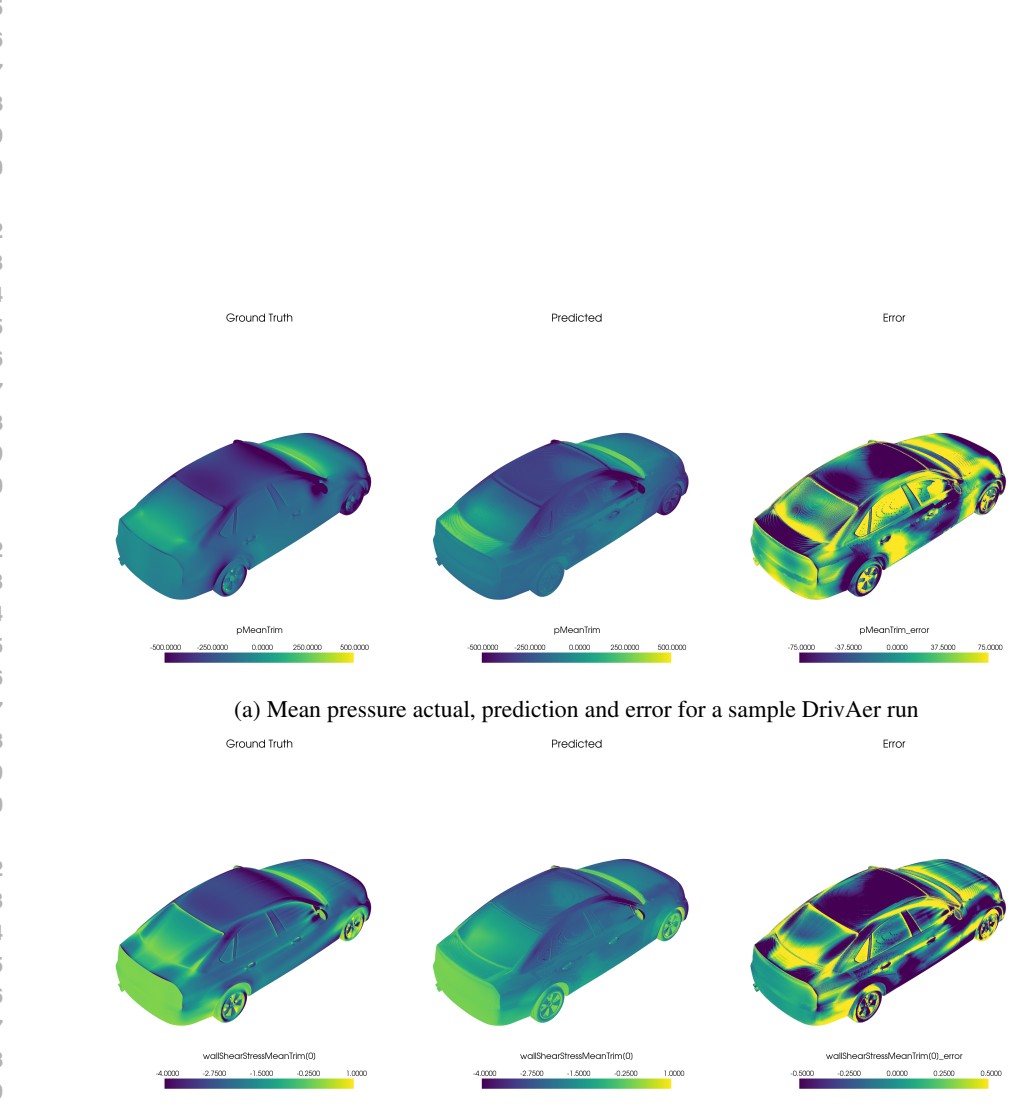

(a) Mean pressure actual, prediction and error for a sample DrivAer run

(b) Mean wall-shear stress actual, prediction and error for a sample DrivAer run

Figure 37: Actual, Prediction and error for the mean pressure and wall-shear stress for a sample DrivAer unseen geometry

# F DATASHEET

## F.1 MOTIVATION

- **For what purpose was the dataset created?** The dataset was created for the development and testing of machine learning methods for Computational Fluid Dynamics and automotive aerodynamics. It addresses current limitations of a lack of high-fidelity training data in this field.

- **Who created the dataset (e.g., which team, research group) and on behalf of which entity (e.g., company, institution, organization)?** The dataset was created by a collaboration between industry, software vendors and cloud computing providers listed in the author list.

- **Who funded the creation of the dataset?** Upstream CFD received partial funding from the German Federal Ministry of Education and Research and the European Union via the EXASIM project (as acknowledged in the main paper). Otherwise, funding was provided in-kind by each of the author organisations (i.e no explicit grant).

## F.2 DISTRIBUTION

- **Will the dataset be distributed to third parties outside of the entity (e.g., company, institution, organization) on behalf of which the dataset was created?** Yes, the dataset is open to the public.

- **How will the dataset will be distributed (e.g., tarball on website, API, GitHub)?** The dataset will be free to download from Amazon S3 (without the need for an AWS account).

- **When will the dataset be distributed?** The dataset is already available to download via Amazon S3.

- **Will the dataset be distributed under a copyright or other intellectual property (IP) license, and/or under applicable terms of use (ToU)?** The dataset is licensed under CC-BY-SA license.

- **Have any third parties imposed IP-based or other restrictions on the data associated with the instances?** No.

- **Do any export controls or other regulatory restrictions apply to the dataset or to individual instances?** No.

## F.3 MAINTENANCE

- **Who will be supporting/hosting/maintaining the dataset?** AWS are hosting the dataset on Amazon S3.

- **How can the owner/curator/manager of the dataset be contacted (e.g., email address)?** The owner/curator/manager of the dataset can be contacted at xxxx (these are also provided in the dataset README and paper).

- **Is there an erratum?** No, but if we find errors we will provide updates to the dataset and note any changes in the dataset README.

- **Will the dataset be updated (e.g., to correct labeling errors, add new instances, delete instances)?** Yes, the dataset will be updated to address errors or provide extra functionality. The README of the dataset will be updated to reflect this.

- **If the dataset relates to people, are there applicable limits on the retention of the data associated with the instances (e.g., were the individuals in question told that their data would be retained for a fixed period of time and then deleted)?** N/A

- **Will older versions of the dataset continue to be supported/hosted/maintained?** Yes, if there are substantial changes or additions, older versions will still be kept.

- **If others want to extend/augment/build on/contribute to the dataset, is there a mechanism for them to do so?** We will consider this on a case by case basis and they can contact xxxxxxx to discuss this further.

2970
2971

### F.4 Composition

2972
2973
2974

- **What do the instances that comprise the dataset represent (e.g., documents, photos, people, countries)?** Each instance represents the aerodynamic field of a different synthetic, generic car geometry of "notchback" type.

2975
2976

- **How many instances are there in total (of each type, if appropriate)?** There are 500 different instances in total.

2977
2978
2979
2980
2981
2982
2983
2984
2985

- **Does the dataset contain all possible instances or is it a sample (not necessarily random) of instances from a larger set? If the dataset is a sample, then what is the larger set? Is the sample representative of the larger set (e.g., geographic coverage)? If so, please describe how this representativeness was validated/verified. If it is not representative of the larger set, please describe why not (e.g., to cover a more diverse range of instances, because instances were withheld or unavailable).** Each instance has been generated synthetically in a parametric fashion. The dataset is a sample of 500 possible instances. The chosen Design Of Experiments algorithm gives a uniform coverage of the sample space, hence the dataset is representative of the larger set by design.

2986
2987
2988
2989

- **What data does each instance consist of? "Raw" data (e.g., unprocessed text or images) or features? In either case, please provide a description.** The "raw" aerodynamic field data, derived quantities (e.g. aerodynamic forces), data reductions (e.g. slices through the volume) and visualisations (image files). Full details are given in Sect. D.

2990
2991

- **Is there a label or target associated with each instance? If so, please provide a description.** Each instance is numerically indexed.

2992
2993
2994
2995

- **Is any information missing from individual instances? If so, please provide a description, explaining why this information is missing (e.g., because it was unavailable). This does not include intentionally removed information, but might include, e.g., redacted text.** No.

2996
2997
2998
2999

- **Are relationships between individual instances made explicit (e.g., users' movie ratings, social network links)? If so, please describe how these relationships are made explicit.** N/A.

3000
3001
3002

- **Are there recommended data splits (e.g., training, development/validation, testing)? If so, please provide a description of these splits, explaining the rationale behind them.** Users are free to decide their own data splits.

3003
3004
3005
3006
3007

- **Are there any errors, sources of noise, or redundancies in the dataset? If so, please provide a description.** Statistical noise is inherent to time-averaged data with finite sample sizes. The statistical error magnitude is quantified for the aerodynamic forces and moments. The run-time of each simulation has been optimised to give a similar statistical error magnitude for drag for all cases.

3008
3009
3010
3011
3012
3013
3014
3015
3016

- **Is the dataset self-contained, or does it link to or otherwise rely on external resources (e.g., websites, tweets, other datasets)? If it links to or relies on external resources, a) are there guarantees that they will exist, and remain constant, over time; b) are there official archival versions of the complete dataset (i.e., including the external resources as they existed at the time the dataset was created); c) are there any restrictions (e.g., licenses, fees) associated with any of the external resources that might apply to a dataset consumer? Please provide descriptions of all external resources and any restrictions associated with them, as well as links or other access points, as appropriate.** The dataset is self-contained.

3017
3018
3019
3020

- **Does the dataset contain data that might be considered confidential (e.g., data that is protected by legal privilege or by doctor–patient confidentiality, data that includes the content of individuals' non-public communications)? If so, please provide a description.** No.

3021
3022

- **Does the dataset contain data that, if viewed directly, might be offensive, insulting, threatening, or might otherwise cause anxiety? If so, please describe why.** No.

3023

If the dataset does not relate to people, you may skip the remaining questions in this section.

- **Does the dataset identify any subpopulations (e.g., by age, gender)? If so, please describe how these subpopulations are identified and provide a description of their respective distributions within the dataset.**

- **Is it possible to identify individuals (i.e., one or more natural persons), either directly or indirectly (i.e., in combination with other data) from the dataset? If so, please describe how.**

- **Does the dataset contain data that might be considered sensitive in any way (e.g., data that reveals race or ethnic origins, sexual orientations, religious beliefs, political opinions or union memberships, or locations; financial or health data; biometric or genetic data; forms of government identification, such as social security numbers; criminal history)? If so, please provide a description.**

### F.5 COLLECTION PROCESS

- **How was the data associated with each instance acquired?** The data was obtained through Computational Fluid Dynamics (CFD) simulations and then post-processed to extract only the required quanitities.

- **What mechanisms or procedures were used to collect the data (e.g., hardware apparatuses or sensors, manual human curation, software programs, software APIs)? How were these mechanisms or procedures validated?** The data was created synthetically using automated software processes. The CFD methodology was validated by comparison to wind tunnel experiment. The processes and their validation is detailed extensively in this document.

- **If the dataset is a sample from a larger set, what was the sampling strategy (e.g., deterministic, probabilistic with specific sampling probabilities)?** The parametric design space has been sampled using a Modified Extensible Lattice Sequence algorithm.

- **Who was involved in the data collection process (e.g., students, crowdworkers, contractors) and how were they compensated (e.g., how much were crowdworkers paid)?** The data generation was carried out by the authors of this document, whose salaries were paid by their employers (stated in the author list).

- **Over what timeframe was the data collected? Does this timeframe match the creation timeframe of the data associated with the instances (e.g., recent crawl of old news articles)? If not, please describe the timeframe in which the data associated with the instances was created.** The data was generated between December 2023 and May 2024.

- **Were any ethical review processes conducted (e.g., by an institutional review board)? If so, please provide a description of these review processes, including the outcomes, as well as a link or other access point to any supporting documentation.** An ethical review process was not considered necessary.

### F.6 PREPROCESSING/CLEANING/LABELING

- **Was any preprocessing/cleaning/labeling of the data done (e.g., discretization or bucketing, tokenization, part-of-speech tagging, SIFT feature extraction, removal of instances, processing of missing values)? If so, please provide a description. If not, you may skip the remaining questions in this section.** No.

- **Was the "raw" data saved in addition to the preprocessed/cleaned/labeled data (e.g., to support unanticipated future uses)? If so, please provide a link or other access point to the "raw" data.**

- **Is the software that was used to preprocess/clean/label the data available? If so, please provide a link or other access point.**

### F.7 USES

- **Has the dataset been used for any tasks already?** Yes, limited testing with various ML approaches has been undertaken by the author team to ensure that the data provided in the dataset is suitable for ML training and inference.

- **Is there a repository that links to any or all papers or systems that use the dataset?** No.
- **What (other) tasks could the dataset be used for?** In addition to ML development and testing, the dataset could be used to explore the flow-physics around a large sample of road-cars and the relationship between them and the resulting aerodynamic force coefficients.
- **Is there anything about the composition of the dataset or the way it was collected and preprocessed/cleaned/labeled that might impact future uses?** Not to the knowledge of the authors.
- **Are there tasks for which the dataset should not be used?** No.

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
