# OpenReview forum: "DrivAerML: High-Fidelity Computational Fluid Dynamics Dataset for Road-Car External Aerodynamics"
_ICLR.cc/2025/Conference — Submitted to ICLR 2025_

### Official Review · Reviewer_EGAc · 2024-10-31

**Soundness:** 3
**Presentation:** 2
**Contribution:** 2
**Rating:** 3
**Confidence:** 4

**Summary:**

This is a solid and valuable contribution. The authors present a large-scale public dataset in the automotive domain and provide detailed information on the dataset generation process. However, the paper reads more like a technical report than an academic paper, and there appears to be a lack of innovative contributions beyond the dataset itself. While the dataset generation is well-documented, adding more baseline testing would strengthen the paper by demonstrating dataset quality and usability, as done in related datasets like PDEBench [1], EAGLE [2], and DrivAerNet++ [3].

[1] PDEBench: An Extensive Benchmark for Scientific Machine Learning

[2] EAGLE: Large-Scale Learning of Turbulent Fluid Dynamics with Mesh Transformers

[3] DrivAerNet++: A Large-Scale Multimodal Car Dataset with Computational Fluid Dynamics Simulations and Deep Learning Benchmarks

**Strengths:**

The authors produced a high-quality looking high-fidelity public dataset from the automotive domain with a solid amount of work.

**Weaknesses:**

1. The paper is written in a style that is more characteristic of a technical report than an academic paper.

2. Lack of comprehensive baseline testing to showcase dataset utility and model performance.

**Questions:**

1. As far as I know, there are already some open-source datasets, for example, DrivAerNet++ [1], what is the difference between your work and their work?

[1] DrivAerNet++:  A Large-Scale Multimodal Car Dataset with Computational Fluid Dynamics Simulations and Deep Learning Benchmarks

2. As a benchmark, how do you evaluate the performance of the model? As a benchmark dataset, it’s important to detail the evaluation metrics and testing procedures used to assess model performance. Consider specifying which metrics were used, how the dataset was split for training/validation/testing, and any comparisons made with baseline methods. This would help highlight how the dataset enables meaningful performance comparisons across models.

3. When defining a "good" public dataset? It would be useful to discuss qualities beyond size alone. For instance, diversity in scenes, real-world relevance, and usability in various model architectures are important factors. Additionally, describing any distribution plans and dataset accessibility options would be valuable in demonstrating the impact and reach of the dataset. This could also increase community engagement and foster broader adoption.

4. What are the dataset requirements for NNs? Since this dataset involves a large number of inputs, guiding model requirements would be useful. Consider outlining preferred model architectures or input-output formats that would be suitable for this data, and discuss any preprocessing or dimensionality reduction techniques recommended for handling the high input volume. This would make it easier for researchers to integrate the dataset into existing models.

---

### Official Review · Reviewer_MyEC · 2024-11-01

**Soundness:** 2
**Presentation:** 2
**Contribution:** 2
**Rating:** 5
**Confidence:** 4

**Summary:**

This paper presents an open-source dataset of high-fidelity computational fluid dynamics (CFD) simulations focused on the aerodynamics of one of the state-of-the-art generic vehicle models DrivAer, with varations in model dimensions. Through the incorporation of morphing boxes this dataset encompasses a wide range of vehicle structures. As a result it presents a valuable asset for developing data-driven approaches of machine learning applications.

**Strengths:**

Accepted challenging benchmark datasets is a current hurdle for the comparison and thus broader usage of SciML. The proposed data set could adress this challenge since it covers a multitude of different vehicle structures due to a large amount of parameters of the morphing boxes. Moreover, a comparison to experimental data from AutoCFD to validate the accuarcy of the simulation is included.

**Weaknesses:**

- In terms of language the paper is generally well written, however there are some imprecise wordings. This includes the choice of geometry that is partly based on "expectations of industrial feasibility" and parameter ranges "based on engineering judgement". More information or a brief discussion is should be added, e.g. to add more concrete criteria for what constitutes "industrial feasibility" or to explain the specific engineering principles that informed their judgment on parameter ranges.

- The missing reference numbers to the corresponding sections in e.g. chapter "Objectives and Main Contributions" and "CFD Methods", as well as the reference to the general appendix (named "SI") instead of explicit chapters throughout the paper impede the understanding of the presented information.

- The low-fidelity dataset "DrivAerNet" is mentioned. Since the datasets aim at similar uasage a more thorough discussion should be provided. In best case a validation that the new dataset can improve data based models. Moreover, a discussion on further related data sets would help to point out the specific benefits of the proposed data set.

- The paper is not self-contained. The appendix contains information that is in fact necessary to undestand the data set and possible usage scenarios. This includes the validation of the method that references the underlying experiment cases and parts of the comparison. This is also the case for the sampling of the parameter space (in the next bullet point). This information is required to understand the data set and potential usage scenarios should thus be included in the paper.

- Even with information from the appendix crucial information for a data set remains unclear. E.g. What is the number of sampled points for each parameter and thus how dense are the parameters sampled? The appendix shows a distribution for two parameters with maybe something between 100 and 200 data points (exakt number not given). Thus there a still 14 dimensions unsampeled, but only 400-500 points left. This seems like a very sparse data set. MeshGraphNets, which promise generalizybility w.r.t geometry, are mentioned. However, sampling seems (from the above estimation) too sparse to achieve this generalization. This should be discussed. Also further potential applications should be discussed w.r.t the size of the data set. Accordingly, the "ML Evaluation" section should be detailed, e.g by
- a complete breakdown of the number of samples for each parameter.
- a discussion on how the sparsity of the dataset might affect its utility for different machine learning tasks, particularly in relation to geometric generalization
- more details on the ML evaluation, including the specific architecture used, loss function formulation, and a more critical analysis of the results in light of the dataset's characteristics.
- the authors thoughts on potential future work to address the dataset's limitations, such as conducting a sensitivity analysis to inform more targeted sampling.

**Questions:**

How does the data set relate to further available data sets in the field of SciML?

Will the data set size be increased in the future?

---

### Official Review · Reviewer_Wopn · 2024-11-02

**Soundness:** 3
**Presentation:** 3
**Contribution:** 3
**Rating:** 8
**Confidence:** 4

**Summary:**

This paper has successfully created a high-fidelity, open-source CFD dataset based on a parametric variation of the DrivAer vehicle model, addressing a critical gap in available training data for ML applications in this domain. The paper thoroughly details the dataset generation process, CFD methodologies, and the potential of the dataset for ML model training and evaluation. The work is well-structured, the results are promising, and the dataset's release under a permissive license is commendable, fostering further research and development. The paper's limitations are acknowledged, and suggestions for future work are provided, indicating a clear path for ongoing research in this area. Overall, the paper is a valuable addition to the literature and would benefit the automotive aerodynamics community by enabling more accurate and efficient design optimization studies.

**Strengths:**

The paper introduces the DrivAerML dataset, which stands out for its originality in several aspects. Firstly, it provides one of the first large-scale, high-fidelity CFD datasets for complex automotive aerodynamics geometries, addressing a significant gap in the availability of open-source training data for ML models in this field. The use of 500 parametrically morphed variants of the DrivAer notchback generic vehicle represents a creative combination of existing ideas, expanding the dataset's applicability beyond a single geometry. This approach not only enhances the diversity of the dataset but also simulates real-world automotive design variations, which is a novel contribution to the field.
The quality of the dataset itself is exceptional, as it is generated using consistent and validated automatic workflows that are representative of industrial state-of-the-art practices. The use of hybrid RANS-LES methods for CFD simulations ensures that the data is of the highest fidelity, which is crucial for the development and testing of accurate ML models. The dataset's comprehensive nature, including full flow-field data, surface data, and application-relevant quantities, further enhances its quality and utility for researchers and practitioners.
The paper is well-structured and clearly articulated. The authors effectively communicate the motivation behind the dataset, its construction, and its potential applications. The clarity of the paper is further enhanced by the detailed descriptions of the CFD methods, the workflow for dataset generation, and the validation against experimental data. The inclusion of visual aids, such as figures and tables, aids in understanding the complexity and diversity of the dataset. Additionally, the paper clearly outlines the structure and contents of the dataset, making it accessible for potential users.
The significance of this paper lies in its potential to revolutionize automotive aerodynamics by enabling faster and more cost-effective fluid flow predictions during the design process. By providing a high-fidelity dataset, the authors empower the research community to develop and test ML models that can significantly accelerate design optimization studies. The dataset's open-source nature and permissive licensing (CC-BY-SA) ensure widespread accessibility and encourage collaborative innovation across academia and industry. The paper's contribution to removing limitations from prior results, such as the lack of high-quality, public-domain CFD data, is significant and has the potential to inspire new research directions and applications in automotive aerodynamics and beyond.

**Weaknesses:**

While the paper provides a comparison of the CFD methodology against experimental data for the baseline geometry, it could benefit from a more extensive validation across a broader range of geometries within the dataset. Actionable Insight: The authors could consider validating the CFD results against experimental data for a subset of the 500 parametrically varied geometries to ensure the dataset's accuracy across different configurations.
The paper mentions the use of statistical quality control in the automated workflows but does not delve into the specifics of data preprocessing steps. Actionable Insight: Providing a detailed account of data preprocessing, including any normalization or filtering applied to the CFD outputs, would enhance the transparency and reproducibility of the dataset.
The dataset focuses on force and moment coefficients, which are crucial, but other aerodynamics metrics such as drag polars or pressure distribution details could provide a more comprehensive understanding of the flow field. Actionable Insight: Expanding the dataset to include a wider array of aerodynamics metrics could increase its utility for researchers interested in specific aspects of vehicle aerodynamics.
The paper conducts a preliminary ML evaluation using a GNN approach but does not explore the performance of other ML models or deeper analysis of model limitations on the dataset. Actionable Insight: The authors could experiment with a variety of ML models and hyperparameter tuning to provide a more thorough evaluation of the dataset's predictive capabilities and to identify any patterns or biases in the data that could affect ML model performance.
Given the rapid advancements in CFD and ML, the dataset might become outdated. Actionable Insight: The authors should consider establishing a protocol for regular updates to the dataset, incorporating new geometries, boundary conditions, and possibly even results from more advanced CFD simulations as they become feasible.
The paper does not discuss plans for community engagement or feedback mechanisms to improve the dataset post-release. Actionable Insight: Establishing a forum or platform for users to provide feedback, suggest improvements, or contribute additional data could foster a collaborative environment around the dataset and enhance its long-term value.
Although the dataset is synthetic, it is derived from real-world automotive design principles. Actionable Insight: The authors might consider an ethical review process to address any potential concerns, even if they are perceived as minor, to set a precedent for responsible data handling in the field.

**Questions:**

The paper mentions the use of statistical quality control but does not detail the data preprocessing steps. Could you provide more transparency on the preprocessing pipeline applied to the CFD outputs?
The dataset primarily includes force and moment coefficients. Are there plans to include additional metrics such as drag polars or detailed pressure distributions in future updates?
The ML evaluation section focuses on a GNN approach. Have you considered the performance of other ML models on this dataset, and what were the outcomes?
Given the rapid evolution of CFD and ML, how do you plan to keep the dataset relevant and up-to-date?
Are there any mechanisms in place for the community to provide feedback or contribute to the dataset post-release?
The paper acknowledges certain limitations. Could you provide a roadmap on how you intend to address these limitations in future work?

---

### Official Review · Reviewer_hHNL · 2024-11-06

**Soundness:** 2
**Presentation:** 2
**Contribution:** 3
**Rating:** 5
**Confidence:** 3

**Summary:**

This paper generates a high-fidelity open-source public dataset, named DrivAerML, for automotive aerodynamics. This dataset consists of 500 variants of the baseline geometry, covering the main features seen on this category of road vehicle. Hybrid RANS-LES is used which is the highest-fidelity scale-resolving CFD approach routinely deployed by the automotive industry, to ensure best possible correlation to experimental data.

**Strengths:**

It is stated that DrivAerML is the first large, open-source ML training dataset comprising high-fidelity CFD data for complex automotive aerodynamics geometries. The dataset primarily targets data-driven surrogate ML approaches. The dataset may also useful for other ML approaches, or even for purposes beyond the ML field. The generation of high-fidelity CFD dataset and the contents and structure of the dataset are described. A supplement material is detailed at the end.

**Weaknesses:**

1.	The section numbers in the paper organization paragraph are missing.
2.	The paper format should be double-checked. The reference format is wrong.
3.	The generation process of the dataset is detailed. However, how does this dataset can be used to facilitate the industrial tasks and academia tasks could be further elaborated. For example, what’s the purpose in Section E.1. The description of the task is not clear. What’s the benefits of this ML evaluation process? What conclusion can we draw from Section E.1 ?
4.	The dataset was created for the development and testing of machine learning methods for Computational Fluid Dynamics and automotive aerodynamics. The baseline ML approaches are not thoroughly tested and listed in the paper. It is recommended to further enrich the baseline models and their corresponding testing performance. These benchmarks and performance info could be hosed on a separated website.

**Questions:**

1.	Baseline geometries are provided at the beginning of the paper, but how these baseline geometries can be further used in other experiments? Whether the size of the dataset is enough for training ML models?
2.	Whether the author could provide more detailed comparison between similar datasets?
3.	It is emphasized in the paper the dataset could be a challenging test-case at future conferences/workshops to benchmark the performance of different ML approaches for an open-source automotive dataset. This dataset is not very common. What’s the so-called test-case for this dataset? How should we define the test-cases? It is not very clear how this dataset is beneficial for academia.

---

> ### Author Response · Authors · 2024-11-29
>
> Thank you very much for the review and apologies for the late reply.
>
> Weaknesses
> For points 1 & 2 we apologize and will fix it in the final paper version. For point 3, we accept that we have assumed that readers of this paper will already be interested in training ML models for CFD and will know about the industrial and academia tasks but I accept we should not assume this and will be sure to add a few more clarifying points on exactly how it could be done.
>
> Point 4 is a crucial one. We purposely have created this paper as very much the ‘data’ part of it, with only minimal ML evaluation to illustrate how it could be used. We believe this still has merit given how detailed the paper is on the dataset generation and validation. We are working on a separate paper with other co-authors to more formally present a detailed ML evaluation.
>
> Questions
> 1: Could you please clarify exactly what you mean by baseline geometries can be further used in other experiments? For the size of the dataset, we have shown in the ML evaluation that using the full 500 cases gives promising accuracy and thus we believe this is a suitable size.
> 2. That is a good point and we will add a section comparing the size/fidelity of the dataset versus others
> 3. the test-case is the DrivAer road-car geometry that is extremely common in the automotive aerodynamics community (for example it is used within the AutoCFD workshop series) and the specification of this test-case is the same in this dataset as that workshop series.

---

### Meta-Review · Area_Chair_uJQp · 2024-12-20

**Metareview:**

This work proposes a dataset for automotive aerodynamics. It aims to support the development of ML-based computational fluid dynamics methods to accelerate flow prediction during the vehicle design process. The main benefit of the work is that it seems to be the first dataset of its type. The reviewers also raised several weaknesses, including the paper's writing, insufficient validation, and lack of additional metrics. These concerns remained largely not addressed during the review process, and thus, this work is not ready for publication.

**Additional Comments On Reviewer Discussion:**

The authors did not participate in the rebuttal except for responding to one review.

---

### Decision · Program_Chairs · 2025-01-22

Reject